# A fluorescent perilipin 2 knock-in mouse model reveals a high abundance of lipid droplets in the developing and adult brain

Sofia Madsen[1], Ana C. Delgado [2], Christelle Cadilhac[3], Vanille Maillard[1], Fabrice Battiston[1], Carla Marie Igelbüscher [1], Simon De Neck [4], Elia Magrinelli [5], Denis Jabaudon [3], Ludovic Telley [5], Fiona Doetsch [2] & Marlen Knobloch [1] ✉

Lipid droplets (LDs) are dynamic lipid storage organelles. They are tightly linked to metabolism and can exert protective functions, making them important players in health and disease. Most LD studies in vivo rely on staining methods, providing only a snapshot. We therefore developed a LD-reporter mouse by labelling the endogenous LD coat protein perilipin 2 (PLIN2) with tdTomato, enabling staining-free fluorescent LD visualisation in living and fixed tissues and cells. Here we validate this model under standard and high-fat diet conditions and demonstrate that LDs are highly abundant in various cell types in the healthy brain, including neurons, astrocytes, ependymal cells, neural stem/progenitor cells and microglia. Furthermore, we also show that LDs are abundant during brain development and can be visualized using live imaging of embryonic slices. Taken together, our tdTom-Plin2 mouse serves as a novel tool to study LDs and their dynamics under both physiological and diseased conditions in all tissues expressing Plin2.

Lipids are essential molecules for cells and organisms and can serve as energy sources, membrane building blocks and signalling entities. However, the nutritional and cellular availability of lipids fluctuates, and excessive lipids can be damaging. Thus, organisms have developed specialized intracellular organelles called lipid droplets (LDs) to buffer and regulate the amount of freely available lipids[1]. LDs contain neutral lipids such as triacylglycerols (TAGs) and cholesterol esters and are surrounded by a monolayer of phospholipids and LD coat proteins. While their primary role is to store and regulate access to lipids, it has become clear that besides their metabolic role, LDs are also involved in other important cellular processes such as regulation of cellular stress[2,3], protein sequestration[4,5], inflammation and immune response[6].

All cells have the capacity to form LDs to a certain extent but under physiological conditions, they are mainly found in lipogenic tissue such as adipose tissue and liver. LDs are tightly linked to lipid metabolism[1], and can upon nutritional excess be newly formed or increase in size to allow for storage of surplus lipids. The liver plays an important role for lipid metabolism, as it both synthesizes and secretes lipids. An imbalance in this process can lead to abnormal accumulation of LDs in the liver, also known as hepatic steatosis[7]. While liver LD accumulation is a well-known side effect of metabolic diseases, LDs can also accumulate in non-lipogenic tissues e.g. upon obesity or cancer[8,9]. LDs also play an important physiological role in certain non-lipogenic tissues, such as muscle, where they can react dynamically to various stimuli[10]. Recent discoveries have put LDs into the spotlight for proper brain functioning[11,12]. Findings in drosophila and mice have linked LD accumulation in the brain to oxidative stress and Alzheimer's disease[13–15]. Furthermore, several other studies have shown that lipids can be transferred from neurons to astrocytes, where astrocytes can

[1]Department of Biomedical Sciences, University of Lausanne, Lausanne, Switzerland. [2]Biozentrum, University of Basel, Basel, Switzerland. [3]Department of Basic Neurosciences, University of Geneva, Geneva, Switzerland. [4]Institute of Veterinary Pathology, University of Zurich, Zurich, Switzerland. [5]Department of Fundamental Neurosciences, University of Lausanne, Lausanne, Switzerland. ✉e-mail: marlen.knobloch@unil.ch

store and subsequently break down lipids. This metabolic coupling between neurons and astrocytes is proposed to protect neurons from lipotoxicity[14,16,17]. Moreover, aging and inflammation are important factors shown to trigger LD accumulation in different brain cell types, including neurons, astrocytes, and microglia[11,18,19], but LDs have also been described in different brain cells under physiological conditions. They are present in the hypothalamus of both mice and human brains[20], as well as in ependymal cells lining the brain ventricles[21,22]. In addition, we recently demonstrated that LDs are important for adult mouse neural stem/progenitor cells (NSPCs)[23].

NSPCs are prominent during early development and persist throughout adulthood, residing mainly in the subventricular zone (SVZ) and dentate gyrus (DG), where they give rise to new-born neurons and glia[24,25]. Both de novo lipogenesis[26,27] and fatty acid beta-oxidation are important pathways for NSPC proliferation and stem cell maintenance and both pathways are closely linked to LD build-up and breakdown[28–31]. We have shown that adult mouse NSPCs in vitro contain variable amounts of LDs, and that an increase in LDs provides a metabolic context that favours their proliferation[23]. Additionally, we found that LD numbers and sizes changed with NSPCs entering quiescence and upon differentiation to neurons and astrocytes[23]. The differences in LD composition between cell types could be an indication that LDs might play different roles within these different cell types. Collectively, these findings have led to an increased interest in understanding the role of LDs in brain cells under both physiological and pathological conditions[11,12,32].

A current limitation in studying LDs in the brain lies in their visualisation: Neutral lipid dyes and antibodies against LD coat proteins allow LD detection in cells, and there are even staining-free microscopy methods to study them in vitro[33,34]. However, these methods have limitations for studying LD dynamics. We therefore developed an LD reporter mouse which allows for staining-free fluorescent visualization of LDs in tissues and cells, both living and fixed. To do so, we endogenously tagged the ubiquitously expressed LD marker and coat protein perilipin 2 (Plin2)[35] with tdTomato using CRISPR/Cas9. We have extensively validated this novel tdTom-Plin2 mouse model under standard and high-fat diet conditions and used it to characterize LDs in the brain.

Using our tdTom-Plin2 mouse, we show that LDs are highly abundant in the brain from embryonic development to adulthood and are detected in the main cell types of the brain including neurons, astrocytes, microglia and oligodendrocytes as well as ependymal cells, endothelial cells, NSPCs and their progeny. We furthermore show that cells can be isolated from the tdTom-Plin2 mice and sorted based on LD content using tdTomato and subsequently used for downstream analysis such as single cell RNA sequencing (scRNA seq). Importantly, we also demonstrate that tissues from these mice can be used for live imaging, enabling the study of LD dynamics in complex set-ups. As Plin2 is ubiquitously expressed, the tdTom-Plin2 mouse will serve as a powerful novel tool to study LDs in different types of tissues during both physiological and pathological conditions.

## Results
### Generation of the endogenous LD reporter mouse and validation of tdTomato-tagged PLIN2 in NSPCs
To visualize LDs in a staining-free manner, we generated a novel LD reporter mouse by tagging the endogenous LD-specific and ubiquitously expressed LD-coat protein PLIN2 with tdTomato. We inserted *TdTomato* with a short linker sequence at the N-terminus of the *Plin2* gene in mouse embryonic stem cells (ESCs) using CRISPR/Cas9. Correctly edited ESCs were injected into mouse blastocysts to generate chimeras, and a germline transmitting tdTom-Plin2 mouse line was established (Fig. 1a). This tagging approach leads to the fluorescent labelling of all PLIN2 positive LDs in all tissues expressing *Plin2* (Fig. 1b). Heterozygous mice were used for all experiments.

We have recently shown that primary NSPCs from adult mice have numerous PLIN2-positive LDs in vitro[23]. We therefore used primary NSPCs, isolated from heterozygous tdTom-Plin2 mice, to validate this novel LD reporter model in vitro. We could detect *TdTomato* mRNA in tdTom-Plin2 NSPCs, and they expressed similar levels of *Plin2* mRNA as control (Ctrl) NSPCs (Fig. 1c). Western blot analysis showed a shift in PLIN2 size due to the heterozygous tdTomato tag in NSPCs from tdTom-Plin2 mice and a reduction in untagged PLIN2 levels compared to Ctrl NSPCs (Fig. 1d). PLIN3, another LD coat protein in NSPCs[23] was not changed (Supplementary Fig. 1a). 95% of the NSPCs expressing PLIN2 also expressed tdTom-PLIN2 and 80% of all LDs expressing PLIN2 showed good colocalization with the tdTom-PLIN2 (Fig. 1e). Quantification of the tdTom-PLIN2 signal in tdTom-Plin2 NSPCs showed a similar area coverage as staining against PLIN2 in Ctrl NSPCs (Fig. 1f).

Having previously shown that LDs influence NSPC proliferation and metabolism[23], we next assessed if the tdTom-PLIN2 tag might alter these two characteristics. EdU pulse-labelling of proliferating cells showed no significant changes in NSPC proliferation in tdTom-Plin2 versus Ctrl NSPCs (Fig. 1g). Using flow cytometry-based cell cycle analysis, we confirmed these results, showing that the tag did not alter NSPC proliferation (Fig. 1h). Furthermore, Ctrl and tdTom-Plin2 NSPCs had similar baseline oxygen consumption (OCR) and extracellular acidification rate (ECAR), suggesting no major alteration in cellular metabolism (Fig. 1i). The ability of the cells to respond to an energy demand (metabolic potential), as shown by an increase in ECAR after inhibition of the ATPase with oligomycin, and an increase in OCR after addition of FCCP, a mitochondrial uncoupling agent, was also similar between Ctrl and tdTom-Plin2 NSPCs, demonstrating that the tag did not alter basic metabolic properties (Fig. 1i).

As tagging a LD coat protein bears the risk of influencing LD build-up, accessibility or turn-over, we next compared the proteome of Ctrl and tdTom-Plin2 NSPCs and found that 99.52% of the proteins were unchanged (Supplementary Fig. 1b and c, Supplementary Data 1). We specifically focused on proteins involved in TAG- and LD build-up, as well as LD breakdown and fatty acid beta-oxidation (FAO). None of those proteins had altered quantities in the tdTom-Plin2 NSPCs compared to the Ctrl NSPCs (Fig. 1j), further showing that the tag does not alter the LD machinery. Of note, PLIN2 was the only perilipin family member detected in sufficient amounts by proteomics, confirming its important role for NSPC LDs. Importantly, PLIN2 protein levels were unaltered (Supplementary Fig. 1d).

Taken together, these data show that the tdTom-PLIN2 signal effectively labels LDs. Importantly the tdTomato tag does not affect the overall expression of *Plin2*, LD abundance, proliferation, basic metabolic properties nor the abundance of proteins involved in LD build-up and breakdown in NSPCs.

### TdTom-Plin2 mice show normal weight gain, normal fat accumulation and label LDs in various organs
Overexpression of *Plin2* has been reported to attenuate lipolysis by reducing the access of adipose triglyceride lipase (ATGL), thus stabilizing LDs[36]. Having endogenously tagged *Plin2*, our reporter system does not rely on overexpression. However, to rule out that the tag on one allele might affect LD stability and fat accumulation, we monitored the weight progression from the age of 3 weeks to 8 weeks of tdTom-Plin2 heterozygous and Ctrl littermates in vivo and found no major differences in weight gain (Fig. 2a). A more detailed fat composition analysis using EchoMRI showed no significant differences in fat mass when comparing 8-week-old tdTom-Plin2 and Ctrl mice (Fig. 2b), thus the tdTomato tag does not seem to influence the body weight and body fat composition of this novel reporter mouse.

*Plin2* is expressed in many tissues such as liver, lung, and heart (Fig. 2c), as shown by a query of the *Tabula muris*, a single cell RNA sequencing (scRNA seq) database of cells from 20 different mouse

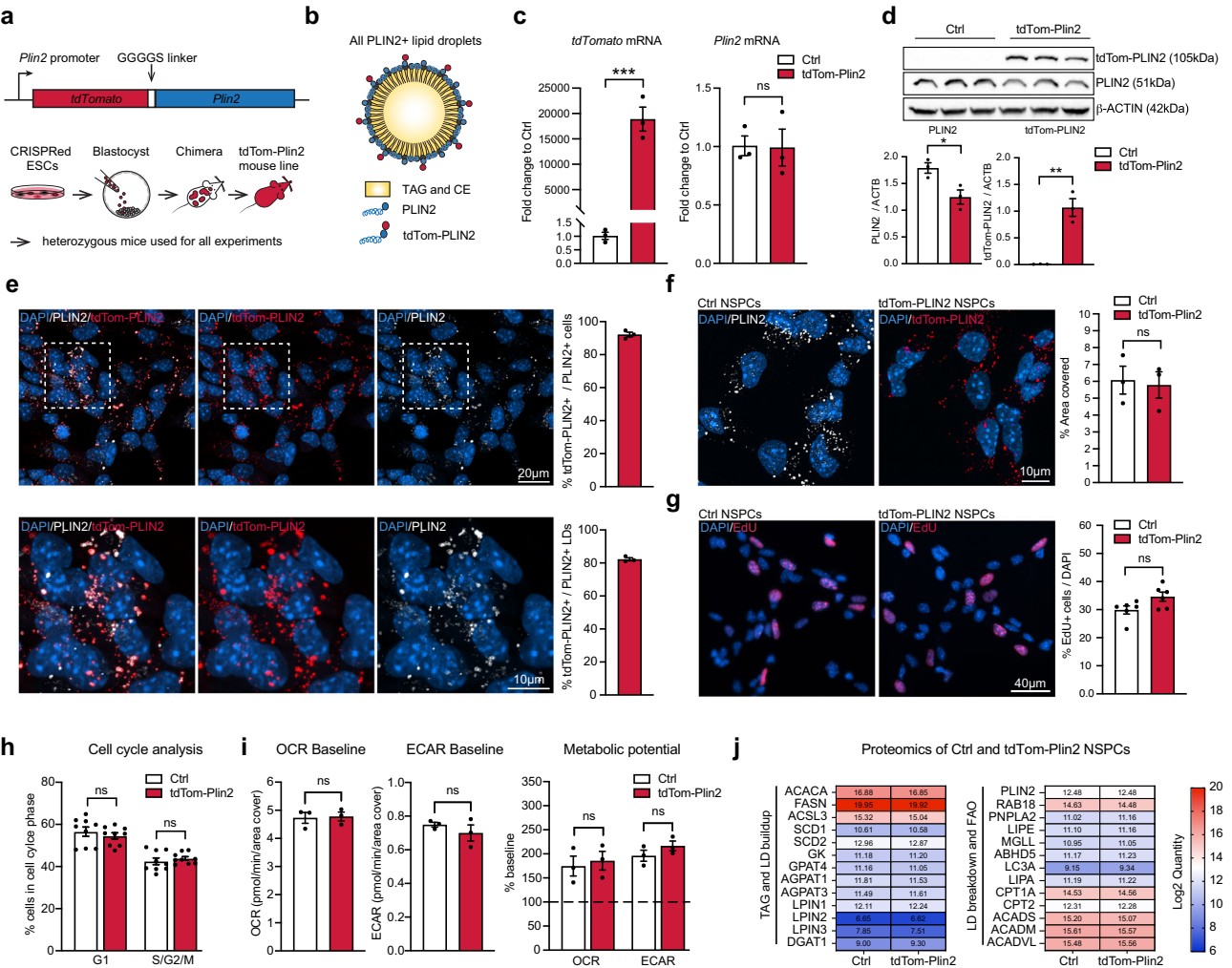

**Fig. 1 | Generation of the endogenous LD reporter mouse and validation of tdTomato-tagged PLIN2 in NSPCs. a** Schematic illustration of the construct and approach used to generate the tdTom-Plin2 reporter mouse. TdTomato is linked to the N-terminus of endogenous *Plin2* through a GGGGS linker. Mouse embryonic stem cells (ESCs) were CRISPRed in vitro. Correctly edited ESC clones were injected into blastocysts to generate chimeras, which were then screened for germ-line expression. Heterozygous mice were used for all experiments. **b** Schematic Illustration for LDs in heterozygous mice, which have LDs coated with wt PLIN2 and tdTomato-tagged PLIN2. **c** Analysis of mRNA expression by RT-qPCR show *tdTomato* expression in NSPCs from the tdTom-Plin2 mouse and no significant difference in *Plin2* between Ctrl NSPCs and tdTom-Plin2 NSPCs. (*n* = 3 samples per condition, fold change +/- SEM. **d** Western blot analysis shows the presence of PLIN2 and tdTom-PLIN2 in NSPCs from the tdTom-Plin2 mice, whereas only untagged PLIN2 is found in Ctrl NSPCs. (*n* = 3 samples per condition, Blots repeated 3 times with similar outcome, mean +/- SEM, uncropped blots in Suppl. Figure 8) **e** Quantification shows that >95% of the NSPCs from the tdTom-Plin2 mouse express tdTom-PLIN2 (*n* = 3 coverslips per condition, 233 cells, mean +/- SEM) and

80% of the PLIN2+ LDs are tdTom-PLIN2+ in NSPCs from the tdTom-Plin2 mouse. (*n* = 3 coverslips per condition, 222 cells, +/- SEM). **f** The area covered by PLIN2 is comparable between Ctrl NSPCs (stained against PLIN2) and tdTom-Plin2 NSPCs (revealed by tdTom-PLIN2). (*n* = 3 coverslips per condition, 237 and 233 cells respectively, mean +/- SEM). **g** Quantification of EdU positive cells shows no significant difference in proliferation between Ctrl and tdTom-Plin2 NSPCs. (*n* = 6 coverslips per condition, mean +/- SEM). **h** Cell cycle analysis by flow cytometry confirms that there is no difference in NSPC proliferation due to the tagged PLIN2. (*n* = 9 samples per condition, mean +/- SEM). **i** Basic metabolic properties are similar between Ctrl and tdTom-Plin2 NSPCs. Bar graphs show baseline oxygen consumption rate (OCR) and extracellular acidification (ECAR), as well as the metabolic potential (*n* = 3 experiments, 2 × 15 and 1 × 14 replicates per experiment and condition, mean +/- SEM) **j** Proteomic analysis of Ctrl and tdTom-Plin2 NSPCs shows no alteration in proteins involved in TAG- and LD build-up, as well as LD breakdown and fatty acid beta-oxidation. Shown are heatmaps of the median values of the log2 quantity (*n* = 4 samples per condition). Asterisks indicate the following *p*-values: * < 0.05; ** < 0.01; *** < 0.001. ns = non-significant.

organs[37]. RT-qPCR confirmed that tdTomato-tagged *Plin2* was expressed in various organs of heterozygous tdTom-Plin2 mice (Fig. 2d). Imaging tdTom-PLIN2 by confocal microscopy furthermore showed clear ring-like signal in tissue sections of liver and intestine from tdTom-Plin2 mice (Fig. 2e). These data show that the endogenously tagged Plin2 is expressed in several organs, thus validating the utility of the tdTom-Plin2 mouse as a versatile tool for studying LDs in various tissue types.

To assess whether any organs of the tdTom-Plin2 mouse shows alterations due to the tag, we performed an extensive histological

evaluation. Three Ctrl and three tdTom-Plin2 adult male littermates were transcardially fixed with PFA and subjected to macroscopic examination and a blinded histological evaluation (Supplementary Fig. 1e). Sections of 40 tissues were microscopically assessed for histological alterations and no differences were found between the two genotypes (Supplementary Fig. 1f). Also, organs known to contain LDs, such as adipose tissue, intestine, heart, muscle and liver did not reveal any differences between Ctrl and tdTom-Plin2 mice (Supplementary Fig. 1g–i), suggesting that the tag is not leading to any abnormal LD accumulation in any of the tissues/organs analyzed.

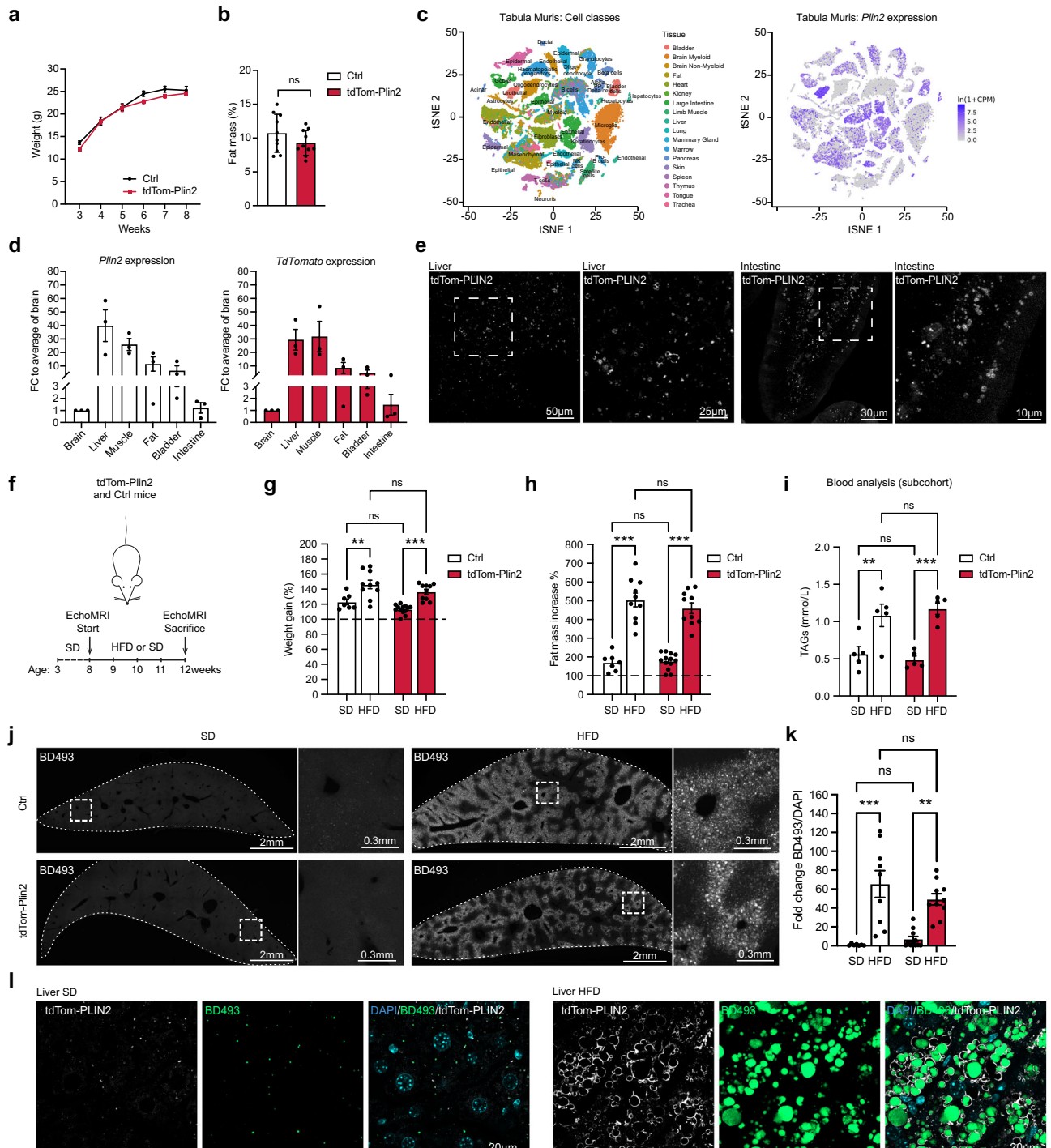

**Fig. 2 | TdTom-Plin2 mice report LDs in several organs and increase both fat mass and LDs in the liver upon a short-term high fat diet. a** There is no major difference in body weight gain over time between Ctrl and tdTom-Plin2 mice in the span of 3- to 8 weeks of age. (*n* = 10 mice per group, mean +/- SEM). **b** Assessment of body fat composition at 8 weeks using EchoMRI shows no significant difference between Ctrl and tdTom-Plin2 mice. (*n* = 10 mice per group, mean +/- SEM). **c** *Plin2* is expressed in many tissues such as liver, lung, and heart, as shown by a query of the *Tabula muris*[37]. **d** *Plin2* and *tdTomato* are expressed in many organs in the tdTom-Plin2 mouse (*n* = 3 mice, mean +/- SEM). **e** Representative images of the tdTomato signal from the liver and the intestine of a tdTom-Plin2 mouse confirm the presence of fluorescently labeled LDs in these organs. **f** Overview of the 4 week high fat diet (HFD) exposure. **g** Both tdTom-Plin2 and Ctrl mice had a significant increase in body weight after 4 weeks of HFD compared to standard diet (SD). (*n* = 7 Ctrl SD, 10 Ctrl HFD, 13 tdTom-Plin2 SD, 10 tdTom-Plin2 HFD mice, mean +/- SEM).

**h** Both tdTom-Plin2 and Ctrl mice had a significant increase in fat mass after 4 weeks of HFD compared to SD. (*n* = : 7 Ctrl SD, 10 Ctrl HFD, 13 tdTom-Plin2 SD, 10 tdTom-Plin2 HFD mice, mean +/- SEM). **i** Serum levels of TAGs were significantly increased in HFD fed animals compared to SD in both tdTom-Plin2 and Ctrl mice. (*n* = 5 mice per group, mean +/- SEM). **j** Representative images of liver sections (single-plane overview and high magnification) stained with BODIPY 493/503 (BD493) of Ctrl and tdTom-Plin2 mice fed a SD or HFD. **k** Quantification of the LD accumulation in the liver, shown in J. Both Ctrl and tdTom-Plin2 mice have a significant increase in LDs upon HFD (BD493 area/DAPI). (*n* = 7 Ctrl SD, 11 tdTom-Plin2 SD, 9 Ctrl HFD and 10 tdTom-Plin2 HFD mice, mean +/- SEM). **l** tdTom-PLIN2 signal also increased significantly in tdTom-Plin2 mice fed a HFD compared to tdTom-Plin2 mice fed a SD, with the tdTomato signal mostly surrounding the BD493 signal. Shown are representative high magnification single plane confocal images. Asterisks indicate the following *p*-values: * < 0.05; ** < 0.01; *** < 0.001. ns = non-significant.

## Exposure to a short-term high fat diet increased fat mass and accumulation of tdTom-PLIN2 in the liver of tdTom-Plin2 mice

We next wanted to assess this novel LD reporter mouse under altered lipid availability. We therefore exposed Ctrl and tdTom-Plin2 mice to a short-term, 4-week high fat diet (HFD), to increase the circulating lipids but limit the manifold side-effects of prolonged HFD such as severe obesity and diabetes. Adult Ctrl and heterozygous tdTom-Plin2 mice were either fed with a HFD containing 60% fat or a standard chow diet (SD) for 4 weeks. We monitored body weight weekly and performed EchoMRI measurements at the beginning and end of the experiment (Fig. 2f). Both Ctrl and tdTom-Plin2 mice on HFD had a 40% weight increase over 4 weeks, whereas the SD fed control groups only had a 10-20% weight gain (Fig. 2g). Similarly, the fat mass increased more than 4 times in both Ctrl and tdTom-Plin2 mice on HFD, whereas SD fed mice only had a 2-fold increase in fat mass over 4 weeks (Fig. 2h). Moreover, we collected serum of a sub-cohort of mice at the end of the experiment to assess how the mice reacted to the diet. Mice fed with HFD had significantly higher serum levels of TAGs (Fig. 2i), cholesterol, LDLs and HDLs (Supplementary Fig. 2a–c), confirming that HFD indeed increased the levels of circulating lipids. We did not observe any differences in free fatty acids (Supplementary Fig. 2d). Blood levels of alanine aminotransferase (ALAT) and aspartate aminotransferase (ASAT), which are used as indicators of liver disease, were not significantly changed with HFD, even though tdTom-Plin2 mice on HFD had a slight elevation in both (Supplementary Fig. 2e, f). Taken together, these data show that tdTom-Plin2 mice respond comparably to increased lipid availability as Ctrl mice, with increased fat mass and weight gain. Thus, tagging *Plin2* does not have an overall effect on lipid storage in the LD reporter mice.

We next examined the livers of Ctrl and tdTom-Plin2 mice, as it has previously been shown that the liver reacts to increased circulating lipids with increased LD numbers and LD sizes[7]. Indeed, quantification of the total area covered by the neutral lipid dye BODIPY493/503 (BD493) showed a significant increase following exposure to a 4-week HFD compared to SD in both Ctrl and tdTom-Plin2 mice (Fig. 2j, k). Similarly, tdTom-PLIN2 also increased significantly in tdTom-Plin2 mice fed with a HFD compared to SD, and mostly surrounded the BD493-positive core of the LDs (Fig. 2l). This increase in LDs was also evident at the protein level of the tagged and untagged PLIN2, as revealed by western blot analysis of liver homogenates from tdTom-Plin2 mice (Supplementary Fig. 2g, h). These data show that lipid availability alters LDs in the liver and that the tdTomato signal faithfully reports such changes.

## LDs are present in multiple cell types in the healthy adult mouse brain

To date, the presence of LDs in the healthy mammalian brain has only been described in a few cell types. As the tdTom-Plin2 mouse reports LDs in isolated NSPCs (Fig. 1) and in the liver and intestine (Fig. 2), we next explored whether the tdTom-Plin2 mouse would be a good tool to visualize and characterize LDs in the postnatal and adult brain. We first measured the expression of *Plin2* in whole brain extracts of 8-week-old tdTom-Plin2 and Ctrl mice. Consistent with our findings from NSPCs in vitro, *tdTomato* mRNA expression was detectable in tdTom-Plin2 mice, while *Plin2* mRNA exhibited a comparable expression pattern to Ctrl brains (Fig. 3a). Expression of the main lipases *Atgl*, *Hsl* and *Mgl* was also comparable between Ctrl and tdTom-Plin2 mice (Supplementary Fig. 3a), although *Atgl* seemed slightly less expressed in tdTom-Plin2 brains. Furthermore, lipidomic analysis showed comparable levels of TAGs, diacylglycerides (DAGs) and monoacylglycerols (MAGs) between brains from tdTom-Plin2 and Ctrl mice (Supplementary Fig. 3b). This suggests that the tdTomato tagged-PLIN2 does not significantly alter neutral lipid composition in the brain.

Confocal microscopy of unstained sagittal brain sections from 8-week-old tdTom-Plin2 mice revealed a surprising abundance and size diversity of fluorescently labelled LDs in various brain regions, such as the olfactory bulb, cortex, the lateral wall of the lateral ventricles and cerebellum (Fig. 3b, c). To validate that this tdTomato signal indeed reports LDs in vivo, we focused on the lateral wall, where studies have reported large LDs in ependymal cells[32]. Distinct tdTomato positive rings of varying sizes were clearly present in the lateral wall confirming their identity as LDs by co-staining with PLIN2 (Fig. 3d).

To elucidate which cell types that contained tdTom-PLIN2 positive LDs under physiological conditions, we crossed tdTom-Plin2 mice to two different GFP-reporter lines (Fig. 3e), labelling neurons (Thy1GFP[38]), and astrocytes (Aldh1l1-GFP[39]), We isolated cells from whole brains and analysed them by FACS (Fig. 3e). Cell-type specific analyses showed that approximately 13% of the Thy1GFP positive neurons had tdTomato positive LDs (Fig. 3f, h and Supplementary Fig. 3c). When examining astrocytes, around 40% of the Aldh1l1GFP positive astrocytes had tdTomato positive LDs detectable by FACS (Fig. 3g, i and Supplementary Fig. 3d).

To further characterise cell types that contain LDs, we used immunohistochemistry to stain for various cell type markers. Imaging of coronal sections of the lateral wall showed tdTomato expression in ependymal cells (S100 Calcium Binding Protein B, S100B), microglia (ionized calcium binding adaptor molecule 1, IBA1), astrocytes (glutamine synthetase, GS), endothelial cells (cluster of differentiation 31, CD31), and oligodendrocytes/oligodendrocyte progenitor cells (oligodendrocyte transcription factor 2, OLIG2) (Fig. 3j). Additionally, we detected LDs in Thy1-GFP positive neurons in the lateral wall of the ventricle in the Thy1GFP tdTom-Plin2 reporter mice. Taken together, these data show that LDs are present in multiple cell types in the mouse brain under physiological conditions and to a much larger extent than previously recognized. This substantial abundance of LDs has not been reported in healthy brain tissue using conventional staining methods employing antibodies against LD coat proteins or lipophilic dyes. We thus explored the reasons for the difference in LDs detected with the tdTom-Plin2 mouse and classical approaches in more detail. We were able to demonstrate that the widely used lipid dye BD493 is ineffective in brain tissue, whereas other commercially available lipid dyes successfully demonstrate the abundance of LDs in wildtype mice, consistent with our observations using the tdTom-Plin2 reporter mouse. Together with a detailed analysis of the effect of detergent and tissue pre-processing on LD detection, these data are presented in a separate manuscript[40].

## ScRNA sequencing of tdTomato-PLIN2 positive microglia reveals a subpopulation with a specific signature under physiological conditions

One important advantage of the tdTom-Plin2 mouse is the staining-free detection of LD-containing cells due to the fluorescence of the tagged PLIN2. This allows for separation of living cell populations based on whether they contain LDs or not and for subsequent characterisation of signature features of such populations. LD-accumulating microglia (LDAM) with a proinflammatory gene expression signature have been identified in the aging brain of mice and humans using BD493 staining[19]. Given that we already detect tdTom-Plin2 positive LDs in microglia in 8-week-old tdTom-Plin2 mice (Fig. 3j), we explored if we could detect a similar signature distinguishing tdTomato positive and tdTomato negative microglia. We used whole brain cell extractions and FACS to separate cells based on tdTomato expression, then performed scRNA seq on the tdTomato positive and negative fractions (Fig. 4a). While we did not specifically preselect the sorted cells for microglia, the majority of good sequencing reads fulfilling the quality criteria mapped to a microglial signature (Supplementary Fig. 4a, b). Thus, we were not able to have an overview of all the brain cell types containing tdTom-PLIN2 positive LDs, but rather focused on the microglial population. Cells with microglial identity clustered into three distinct clusters when using unsupervised

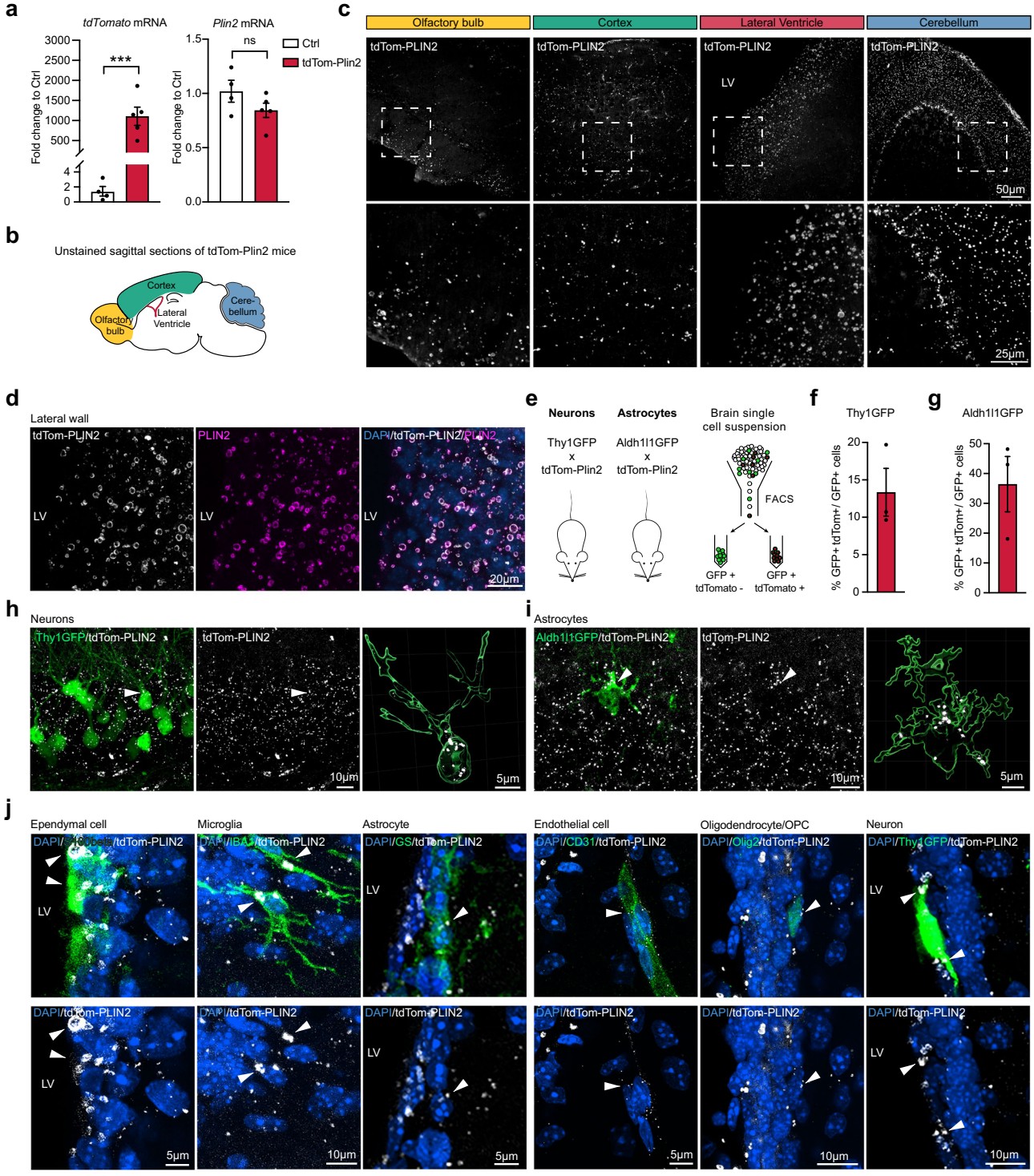

clustering (Fig. 4b). Interestingly, the three clusters contained distinct proportions of tdTomato positive cells (Fig. 4c). Cluster 0 contained comparable numbers of tdTomato positive and tdTomato negative cells (56% vs 44%), cluster 1 had lower proportion of tdTomato positive cells (24% vs 76%) whilst cluster 2 was almost entirely composed of tdTomato positive cells (95% to 5%) (Fig. 4d). The *Plin2* expression matched this distribution, validating the sorting approach (Fig. 4e). A heatmap representation based on the top differentially expressed genes per cluster showed a clear separation of the tdTomato positive cluster 2, whereas cluster 1 and 0 were more similar (Fig. 4f). To better understand the cluster 2 specific signature, we compared differentially

expressed genes between cluster 1 (mainly tdTomato negative cells) and cluster 2 (mainly tdTomato positive cells). 395 genes were significantly upregulated in the tdTomato positive cluster 2, and 153 significantly downregulated (Fig. 4g, Supplementary Data 2). A gene ontology (GO) analysis of the upregulated genes revealed the enrichment of several terms involved in cellular activation (e.g. GO: 0006412: translation; GO:0050865: regulation of cell activation) as well as in immune functions (e.g. GO:0001817: regulation of cytokine production; GO:0002252: immune effector process; GO:0001774: microglial cell activation). Together, these results suggest that tdTomato positive microglia might represent a fraction of microglia undergoing

**Fig. 3 | LDs are present in multiple cell types in the healthy adult mouse brain.**
**a** *TdTomato* expression is clearly detectable in tdTom-Plin2 brains, and *Plin2* mRNA is expressed in a similar manner as in Ctrl brains. (*n* = 4 Ctrl and 5 tdTom-Plin2 mice, fold change +/- SEM). **b** Schematic illustration of a sagittal brain section, highlighting specific regions **c** tdTom-PLIN2 signal in the olfactory bulb, cortex, lateral ventricle well and cerebellum shows abundant LDs in the brain. Representative maximum intensity projections at low (top) and high (bottom) magnifications. (*n* = 3 mice, similar observations in all the mice) **d** TdTomato positive rings of various sizes are clearly present in the lateral ventricle wall and colocalize with the signal from staining against PLIN2, confirming that they are LDs. Representative maximum intensity projection of individual channels and a merge. **e** Scheme of the crossing of tdTom-Plin2 mice to Thy1GFP reporter mice, labelling neurons, and Aldh1l1GFP reporter mice, labelling astrocytes, and scheme of the FACS procedure to determine the number of GFP+ tdTomato+ cells. **f** FACS analysis of whole brain cell extraction from the tdTom-Plin2 mice crossed to Thy1GFP mice show 13% of all Thy1GFP positive neurons have LDs. (*n* = 3 mice, mean +/- SEM). **g** FACS analysis of whole brain cell extraction from the tdTom-Plin2 mice crossed to Aldh1l1GFP mice show that around 40% of all Aldh1l1GFP positive astrocytes have LDs. (*n* = 3 mice, mean +/- SEM). **h** Representative confocal image of an unstained hippocampal section. Thy1-GFP neurons (green) and tdTom-Plin2 signal (white), maximum projection of a z-stack over the volume of the neuron indicated. Right panel shows a 3D rendering to visualize LDs inside the indicated Thy1-GFP neurons. **i** Representative confocal image of an unstained hippocampal section. Aldh1l1-GFP astrocyte (green) and tdTom-Plin2 signal (white), maximum projection of a z-stack over the volume of the astrocyte). Right panel shows a 3D rendering to visualize LDs inside the indicated Aldh1l1-GFP astrocyte. **j** Immunohistochemical staining of the lateral wall show a large variability of endogenous tdTom-PLIN2 positive LDs in S100b-positive ependymal cells, IBA1-positive microglia, Glutamine synthase (GS)-positive astrocytes, CD31 positive endothelial cells, OLIG2-positive oligodendrocytes/oligodendrocyte progenitors and a Thy1GFP neuron. Representative maximum intensity projections of observed cell types (z-projections over the volume of the cells indicated by arrows). (*n* = 3 mice, similar observations in at least 10 cells per cell type and mouse). Asterisks indicate the following *p*-values: *** < 0.001. ns = non-significant.

activation. Previous studies have identified microglial signatures related to disease states (disease-associated microglia, DAM) and a recent review summarized the most common microglial gene expression alterations across various neurodegenerative diseases[41]. We queried whether these genes differed between our clusters 1 and 2. Indeed, many genes upregulated with disease also had a higher expression in tdTomato positive microglia (Supplementary Fig. 4c). However, so-called homeostatic microglial genes which are usually downregulated with disease, were also higher expressed in the tdTomato positive microglia (Supplementary Fig. 4c), suggesting that these microglia might be prone to adapt a DAM-like signature, but are not yet in the same state. Similarly, the comparison with the most significantly up- and downregulated LDAM genes[19] showed that tdTomato positive microglia share some of the LDAM features but are not entirely similar (Supplementary Fig. 4d). Whether microglia will shift their gene expression profile with aging or disease remains to be determined.

Taken together, we demonstrate the utility of the tdTom-Plin2 mouse in identifying and characterizing specific cell subpopulations based on their LD content. Notably, we present here a proof-of-concept with microglia, however, this approach should be applicable to any cell type than can be isolated and subsequently sorted by FACS.

## LDs are present in NSPCs and their progeny in the postnatal and adult mouse brain

Next, we aimed to characterise LDs in NSPCs in vivo by crossing tdTom-Plin2 mice with Nestin-GFP mice (Nes-GPF)[42]. Nes-GFP labels the NSPCs in the dentate gyrus (DG) and the NSPCs and ependymal cells in the subventricular zone (SVZ) (Fig. 5a). TdTomato positive LDs were detected in both the DG and the SVZ in brains of 1-, 3- and 8-week-old mice (Fig. 5b, c, Supplementary Fig. 5a, b). We performed FACS analysis of micro-dissected SVZ and hippocampus of 8-week-old NesGFP-mice to assess the numbers of double positive cells. Around 30% of the NesGFP positive NSPCs in the hippocampus were tdTomato positive (Fig. 5d and Supplementary Fig. 5c) respectively 13% of the NesGFP positive cells in the SVZ were tdTomato positive (Fig. 5e and Supplementary Fig. 5d). However, since NesGFP is also partially expressed in ependymal cells[42], we cannot define the exact percentage of NSPCs in the SVZ that have LDs. In summary, we show that NSPCs in vivo have LDs, but not to the same extent as in vitro where almost all NSPCs have LDs (Fig. 1e)[23].

To further characterize LDs in NSPCs and their progeny, we stained for different markers of the NSPC lineage in the SVZ. We observed LDs in some, but not all, glial fibrillary acidic protein (GFAP) expressing radial stem cells, epidermal growth factor receptor (EGFR) expressing transit amplifying cells, doublecortin (DCX) expressing neuroblast and neuron-glial antigen 2 (NG2) expressing oligodendrocyte progenitors (Fig. 5f).

Taken together, our endogenous LD reporter mouse reveals the presence of LDs in both NSPCs and their progeny in the adult mouse brain under physiological conditions.

## A short-term 4-week HFD slightly increases the total TAG levels in the brain and moderately increases LDs in the wall of the lateral ventricles

We have previously shown that supplementation with exogenous lipids in vitro increases LDs in NSPCs[23]. Additionally, Hamilton and colleagues showed that infusion of oleic acid (OA) directly into the lateral ventricle of adult mice using an osmotic pump led to an increase of LDs in the ependymal layer of the SVZ[15]. We therefore investigated if a short term HFD affects LDs in the mouse brain. We first assessed gene expression of *Plin2* and *tdTomato* in brains from Ctrl and tdTom-Plin2 mice which had been fed either with a SD or short term HFD (Fig. 2e). Expression of neither *Plin2* nor *tdTomato* were significantly changed due to the diet (Fig. 6a, b), suggesting that *Plin2* expression does not increase in response to a short term HFD in the brain. However, changes in LDs may occur locally, and therefore difficult to detect at the global mRNA level from the whole brain. As LDs mainly store TAGs, we performed quantitative lipidomics on the same brains, a highly sensitive method providing a detailed analysis of the neutral lipid composition in the brain. The lipidomics revealed that HFD fed mice had a slight but significant increase in TAGs, both in the Ctrl mice and in the tdTom-Plin2 mice (Fig. 6c). This became even more evident when examining the TAG levels independently of genotype (Fig. 6d). The precursors of TAGs; DAGs and MAGs, were not significantly changed in any of the experimental groups (Supplementary Fig. 6a, b). A more detailed analysis of the lipid composition in the brains of HFD compared to SD fed mice showed that 18% of the measured lipids were significantly changed in either Ctrl or tdTom-Plin2 mice, or in both groups (Fig. 6e). Notably, the majority of the significantly changed brain lipids were TAGs (Fig. 6f). When looking at the fold changes of the significantly changed lipids, both Ctrl and tdTom-Plin2 brains showed a coherent pattern in terms of increased and decreased lipid species as a reaction to the HFD (Fig. 6f).

We next assessed if the increase in TAGs was evident on the LD level. We used wholemounts of the lateral ventricles of heterozygous tdTom-Plin2 mice fed either a SD or HFD and analysed the area covered by the tdTom-PLIN2 signal. Wholemounts provide a clear visualisation of LDs in ependymal cells and NSPCs in both SD and HFD mice (Supplementary Fig. 6d). Segmentation of the wholemounts into several rostral and medial parts of the lateral wall, followed by quantification of the area covered by tdTom-PLIN2, revealed a slight increase in LDs, particularly in the rostral parts, in mice on the HFD (Fig. 6g, Supplementary Fig. 6c, e). However, the increase was relatively moderate and only showed a statistical trend (*p* = 0.0509).

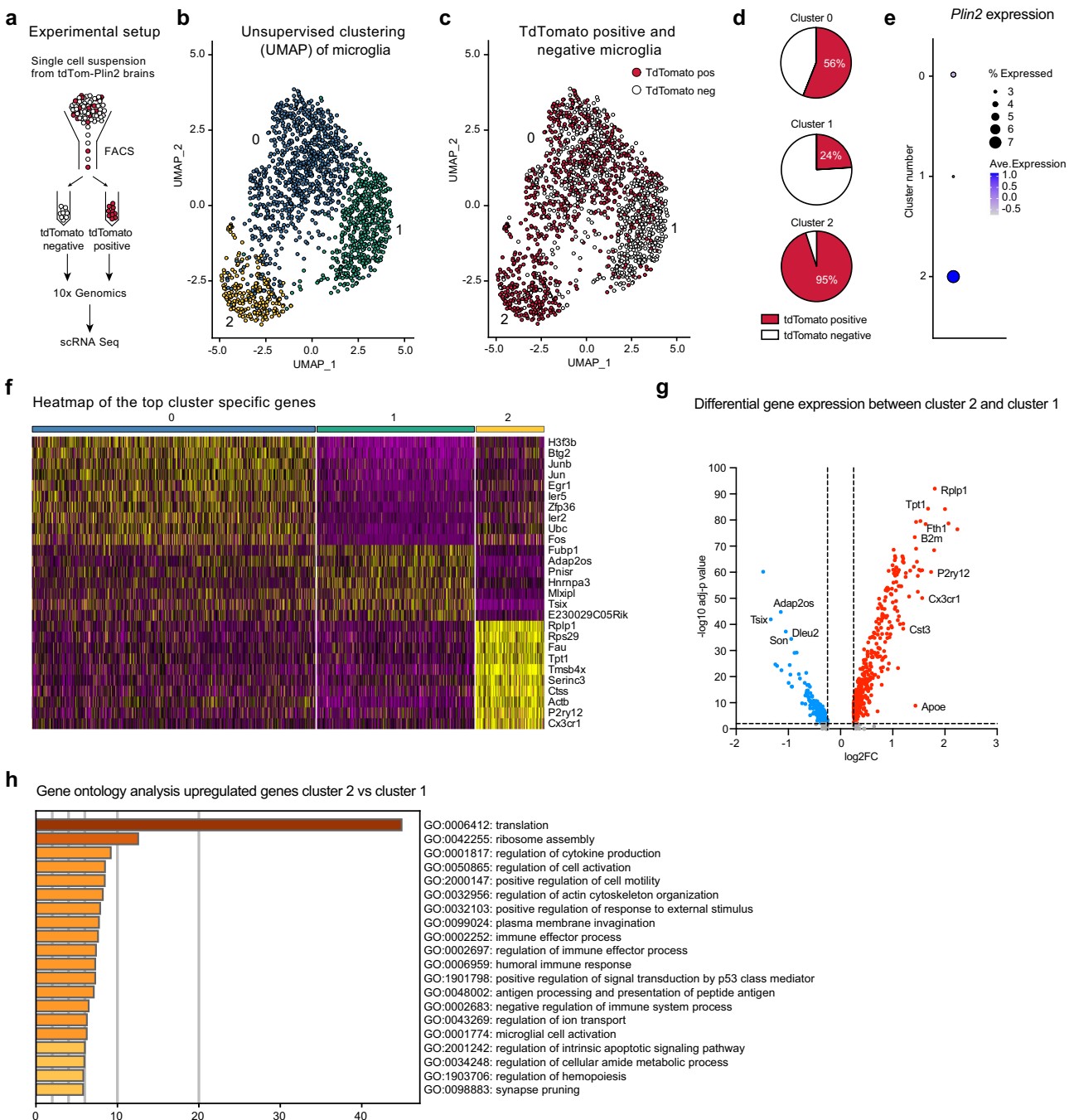

**Fig. 4 | ScRNA sequencing of tdTomato-PLIN2 positive microglia reveals a subpopulation with a specific signature under physiological conditions.**
**a** Schematic illustration of the experimental setup. **b** Unsupervised clustering of cells with microglial identity reveals three distinct clusters. **c** The three microglial clusters contain distinct proportions of tdTomato positive cells. **d** Quantification of tdTomato-PLIN2 positive and negative cells per cluster identifies cluster 2 as highly enriched for microglia containing LDs, while cluster 1 contains a majority of tdTomato negative cells. **e** The *Plin2* expression matches this distribution, validating the sorting approach. **f** A heatmap of the top cluster specific genes shows a clear separation of the tdTomato-PLIN2 positive cluster 2, whereas cluster 1 and 0 had a less striking difference in gene expression. **g** Volcano plot of differentially expressed genes between cluster 2 and cluster 1. **h** A gene ontology (GO) analysis of the upregulated genes in cluster 2 reveals the enrichment of several terms involved in cellular activation as well as in immune functions.

Taken together, we show that the brain possesses a limited capacity to store TAGs, which slightly increases upon a short term 4-week HFD. Moreover, we found that the increase in TAGs was accompanied by a moderate increase in tdTom-PLIN2 positive LDs in the lateral ventricle walls. This indicates that, at least in ependymal cells, LD accumulation responds to heightened lipid availability. Whether other brain regions also react with increased LDs remains to be determined.

## LDs are present in the developing embryonic brain and react dynamically to exogenous lipids

Little is known about the role of lipid metabolism and LDs during embryonic brain development. As both have been shown to play a role in the regulation of adult NSPCs[31], we investigated if LDs were also present in embryonic brains where NSPCs are actively generating new cells at a high rate. Querying existing scRNA seq data from the developing brain (http://mousebrain.org)[43], we saw that *Plin2* was

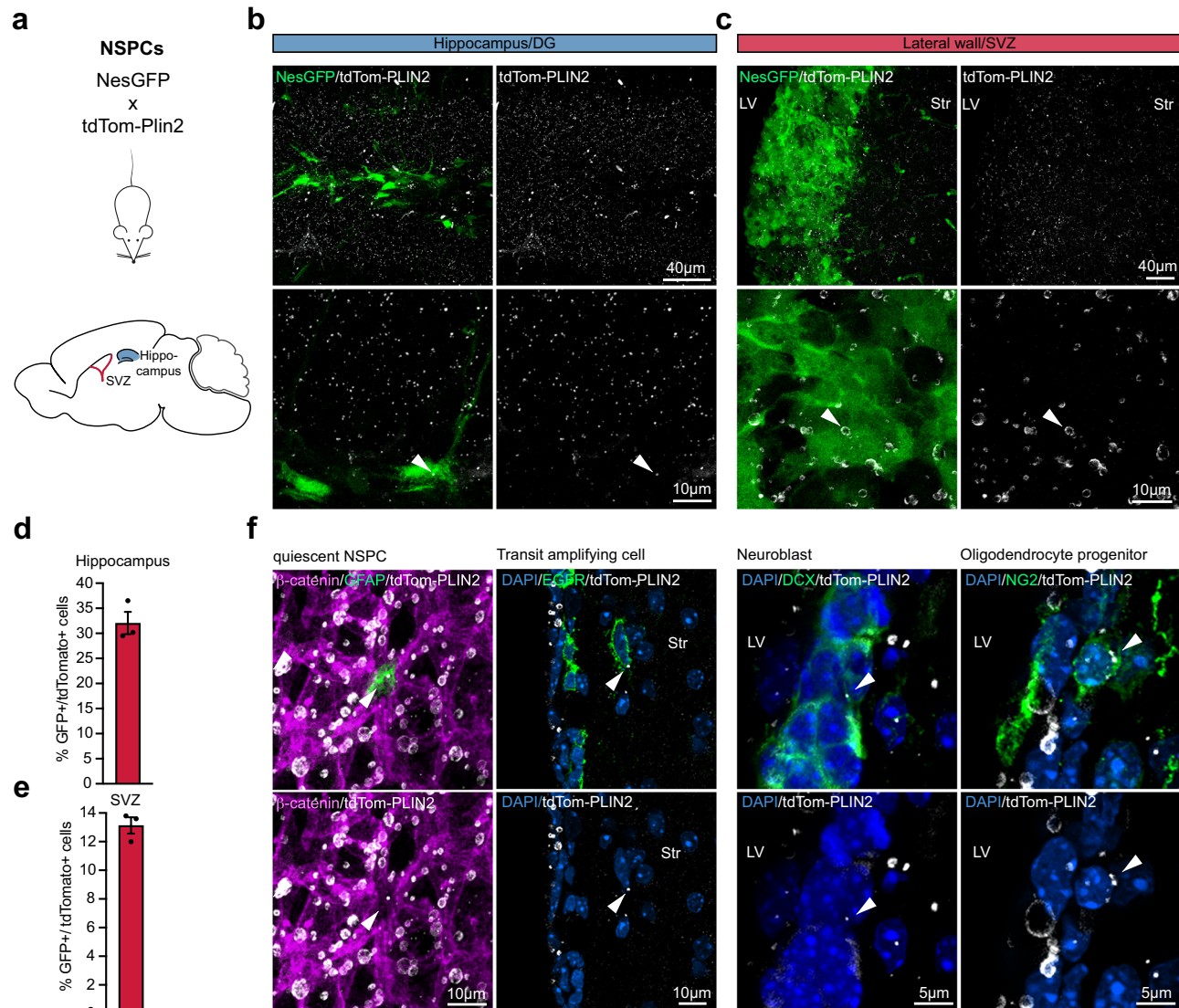

**Fig. 5 | LDs are present in NSPCs and their progeny in the postnatal and adult mouse brain. a** Scheme of the crossing of tdTom-Plin2 mice to NesGFP reporter mice, labelling NSPCs in the SVZ and DG. **b**, **c** Endogenous tdTom-PLIN2 signal in the DG and SVZ showing LDs in NSPCs. Representative images of non-stained sections show NesGFP positive NSPCs and tdTom-PLIN2 positive LDs. (Maximum intensity projections,10 μm stacks). **d** FACS analysis of cells isolated from the hippocampus of tdTom-Plin2 mice crossed to NesGFP mice show that around 30% of the NesGFP positive NSPCs have LDs. (n = 3 mice, mean +/- SEM). **e** FACS analysis

of cells isolated from the SVZ of tdTom-Plin2 mice crossed to NesGFP mice show that around 13% of the NesGFP positive NSPCs have LDs. (n = 3 mice, mean +/- SEM). **f** Immunohistochemical staining of cells in the SVZ show tdTom-PLIN2 positive LDs in GFAP positive quiescent NSPCs (whole mount), EGFR positive transit amplifying cells, DCX positive neuroblasts and NG2 positive oligodendrocyte progenitors (coronal sections). Representative maximum intensity projections of the observed cell types. (n = 3 mice, similar observations in at least 10 cells per cell type and mouse).

expressed in several cell type clusters (Fig. 7a). Embryonic NSPCs, also called radial glial cells (RG) were one of the clusters with more abundant *Plin2* expression (Fig. 7a). Moreover, neuroblasts and neurons also expressed *Plin2*. In line with these scRNA seq data, we found a wide distribution of LDs across the entire thickness of the developing cortex in unstained brain sections from tdTom-Plin2 embryos at E14.5, a stage of highly active neurogenesis (Fig. 7b, c). We confirmed by immunohistochemical staining that the tdTomato signal colocalized well with PLIN2 (Supplementary Fig. 7a). We next looked in more detail at the ventricular/subventricular zone, which can be labelled with the marker Sox2, and where RGs and intermediate progenitors reside. TdTomato intensity in relation to distance from the ventricle was similar throughout the Sox2-positive layer (Fig. 7d), suggesting a homogenous LD distribution across the ventricular/subventricular zone.

To investigate how LDs respond to exogenous lipids, we incubated acute embryonic brain slices with 0.1 μM oleic acid (OA) overnight. This led to a significant increase in the area covered by tdTom-PLIN2 positive LDs (Fig. 7e). Additionally, we investigated the dynamics of the LDs by live-imaging brain slices from E14.5 tdTom-Plin2 embryos for 2 h under two conditions: i) sections imaged in artificial cerebrospinal fluid (aCSF), ii) sections imaged in aCSF with 0.1 μM OA after overnight incubation in medium containing 0.1 μM OA (Supplementary Fig. 7b). Surprisingly, LDs decreased continuously in the nontreated condition during imaging (Supplementary Fig. 7c), while the decrease in LDs was less prominent after overnight incubation with OA (Supplementary Fig. 7d). As aCSF contains less nutrients than the medium in which the sections are pre-incubated in, these results suggest that cells utilize the lipids stored in LDs over time if there are not enough exogenous lipids provided. As photobleaching could have

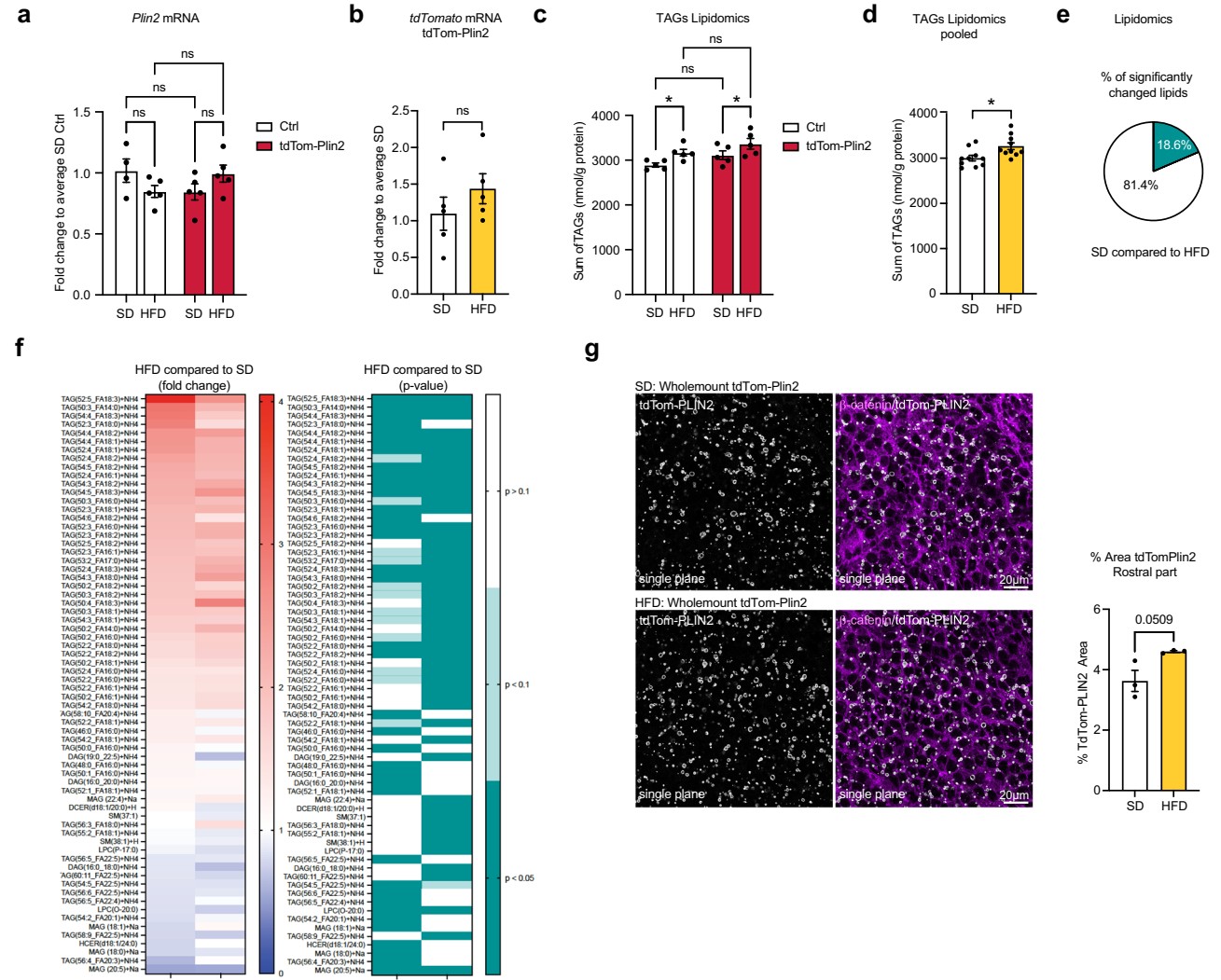

**Fig. 6 | Short-term high fat diet slightly increases the total TAG levels in the brain and moderately increases LDs in the wall of the lateral ventricles. a** Analysis of mRNA expression by RT-qPCR shows no significant difference in *Plin2* due to the HFD in either Ctrl mice or tdTom-Plin2 mice. ($n = 4$ Ctrl SD, 5 Ctrl HFD, 5 tdTom-Plin2 SD, 5 tdTom-Plin2 HFD mice), fold change +/- SEM). **b** *tdTomato* was also not altered on the gene expression level in SD or HFD tdTom-Plin2 mice ($n = 5$ mice per group, fold change +/- SEM). **c** Lipidomic analysis revealed an increase in brain TAGs upon HFD compared to SD, ($n = 5$ mice per group, mean +/- SEM). **d** Pooling of genotypes shows that TAGs are significantly increased in HFD fed mice compared to SD mice, however the increase is overall very moderate. ($n = 10$ mice per group, mean +/- SEM). **e** Out of 366 analysed lipids in Ctrl and tdTom-Plin2

brains by lipidomics, 18.6% were significantly different when comparing SD to HFD. **f** The majority of the significantly changed brain lipids were TAGs. When looking at their fold changes, both Ctrl and tdTom-Plin2 brains showed a coherent pattern in terms of increased and decreased lipid species as a reaction to the HFD (left heatmap shows fold change to SD, right heatmap shows the corresponding *p*-values). **g** Quantification of the area covered by tdTom-Plin2 signal in whole mount preparations of the lateral ventricle wall shows a slight increase in LDs with HFD in the rostral part. Representative single planes, β-catenin outlines the cell membranes of ependymal cells. ($n = 3$ mice per group, mean +/- SEM). Asterisks indicate the following *p*-values: *< 0.05. ns = non-significant.

---

contributed to the effect, we repeated the imaging, reducing the interval of pictures to every 30 min instead of 5 min and observed a similar decrease, suggesting that photobleaching is not the major cause of the decrease (Supplementary Fig. 7e).

In conclusion, we show that LDs are highly abundant in the developing mouse cortex, and that the tdTom-Plin2 mouse also allows visualization of LDs during embryonic brain development. Furthermore, our results indicate that LDs react dynamically to exogenous lipids. As a proof-of-concept, live-imaging illustrates the usefulness of the tdTom-Plin2 reporter mouse to study LDs in a dynamic and staining-free manner.

## Discussion

LDs have recently gained new interest as regulated and dynamic organelles, with important functions in both health and disease. Here

we describe a novel LD reporter mouse, generated by labelling the endogenous LD specific protein PLIN2 with tdTomato using CRISPR/Cas9. We moreover demonstrate the usefulness of this tdTom-Plin2 mouse with various techniques, including staining-free detection, scRNA seq, and live imaging. We validate the tdTom-Plin2 mouse model under SD and short-term HFD and demonstrate that an increase in LDs is detectable in the livers of the HFD mice. We have further used this mouse model to study LDs in the brain where we show an unexpected abundance of LDs in multiple brain regions and cell types in healthy conditions. We could also show that brain TAGs and LDs slightly increase already after a 4-week HFD. Furthermore, we demonstrate that LDs are present during embryonic brain development and can dynamically react to exogenous lipids, opening new questions on the effect of dietary lipids on brain development and function.

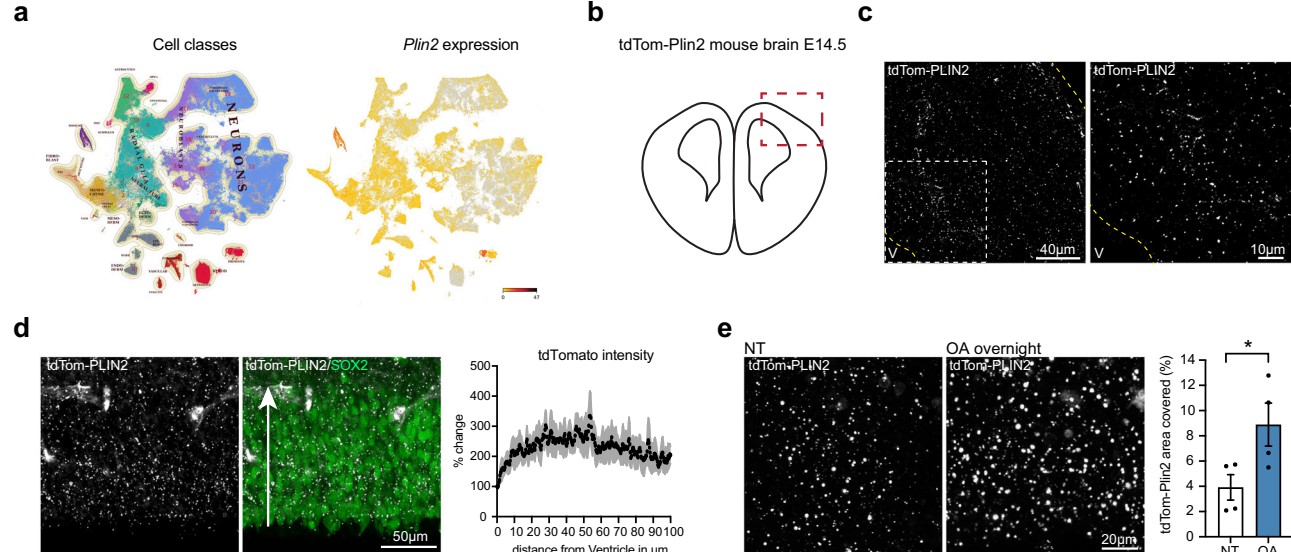

**Fig. 7 | LDs are present in the developing embryonic brain and react dynamically to exogenous lipids. a** Querying for *Plin2* expression in scRNA sequencing data from embryonic brain development (http://mousebrain.org) shows that *Plin2* is abundant in many clusters, one of them being radial glia (RG) cells. **b** Illustration of a coronal section from an embryonic brain at E14.5, red square marking the dorsal pallium, where LDs were imaged. **c** tdTom-PLIN2 positive LDs are abundant in the developing cortex at E14.5 in tdTom-Plin2 embryos. (Maximum intensity projection, 25 μm stack, at low and high magnification). **d** Quantification of tdTomato intensity in relation to distance from the ventricle shows that LDs are homogenously distributed throughout the Sox2-positive layer. Arrow indicates the direction of measurement (*n* = 3 embryos, mean +/- SEM). **e** Overnight incubation of OA significantly increases the LD area covered compared to non-treated (NT) sections (*n* = 4 embryos, mean +/- SEM). *< 0.05.

Several tools are currently available for studying LDs, including classical immunofluorescent staining, lipid dyes such as OilRedO and BD493 and recently developed probes based on lipophilic fluorophores. As they either require fixation prior to staining, incubation prior to imaging or have limitations in being specific for LDs[33,44,45], better models are needed to study LD dynamics. Existing staining-free approaches comprise specific microscopy techniques, such as Raman microscopy[46] or transgenic LD reporters[47], and a few transgenic invertebrate model organisms, such as C.elegans[48] and Drosophila[49]. A very recent addition to the LD imaging toolbox are two LD reporter zebrafish models, one with a fluorescently tagged Plin2 transgene[50] and the other one with fluorescent knock in Plin2/Plin3[51]. However, a mammalian LD reporter model has been missing to date. Thus, our mouse model adds a valuable tool to the field of LD biology and can be of wide use, as it labels LDs in all cells and tissues expressing Plin2.

Tagging proteins with large fluorophores such as tdTomato always bears the risk of altering the expression or function of the tagged protein. To minimize the impact of the tag, we chose a heterozygous tagging strategy, leaving one *Plin2* allele unaltered. Here we show that tagging of endogenous *Plin2* with tdTomato does not significantly alter the mRNA expression, protein expression nor the amount of LDs in NSPCs in vitro. We further show that endogenous tdTom-PLIN2 colocalizes well with stained PLIN2 both in cultured NSPCs and in the brain. Moreover, we report no changes in weight and fat composition of the tdTom-Plin2 mouse under SD and HFD compared to control mice, indicating that the tag does not result in global physiological alterations. This is supported by an extensive histopathological analysis of tdTom-Plin2 and Ctrl littermates, which did not reveal any obvious difference in 40 tissues analyzed. Our brain lipidomics analysis further suggests that tdTomato-tagged PLIN2 does not lead to an accumulation or decrease of neutral lipids in the brain, as TAGs, DAGs and MAGs were similar between control and tdTom-Plin2 mice. Furthermore, HFD lead to very similar changes in the brain lipidome in both tdTom-Plin2 and control mice. However, we cannot entirely rule out the possibility that the tag may induce subtle changes in LD- or PLIN2 turnover, as it could influence access to LDs or protein degradation. This might be to a certain extend tissue- or cell specific and needs to be kept in mind.

Using the tdTom-Plin2 mouse, we show that LDs are found in multiple cell types of the brain under physiological conditions in a high abundance not previously reported. The discrepancy of our findings with the tdTom-Plin2 reporter mouse, compared to classical LD staining approaches, has prompted us to further investigate why LDs are not seen to the same extent in healthy wildtype mouse brains. Our investigations show that the discrepancy arises from technical challenges in visualizing LDs. Specifically, we discovered that the use of specific lipophilic dyes and optimized protocols enables detection of a similar extent of LD accumulation in the healthy adult wildtype mouse brain[40]. LDs are fragile organelles; thus detergent use can influence their detection. Furthermore, the very lipid-rich myelin in the brain might influence the performance of lipophilic dyes, therefore the same dyes which are able to detect LDs in other tissues might not work to the same extend in brain tissue. We were indeed able to demonstrate that the most used dye, BD493, does not work well in brain tissue[40]. Thus, special care and adapted protocols must be used when studying LDs in the brain using classical staining approaches.

The advantage of the tdTom-Plin2 mouse becomes evident when one wants to examine the LD content of cells using FACS. Crossing this model to any fluorescent GFP reporter line allows the selective sorting of GFP positive cells containing PLIN2-positive LDs, eliminating the need for staining. Indeed, crossing the tdTom-Plin2 mouse with neuron- and astrocyte specific GFP reporter lines allowed us to further characterize that around 13% of all Thy1GFP positive neurons, and almost half of all the Aldh1l1GFP positive astrocytes at a given timepoint have PLIN2-containing LDs. It is worth noting that the Thy1GFP mouse sparsely labels neurons throughout the brain[38], and Aldh1l1GFP, although specific to astrocytes, might not be expressed in all astrocytes. As we used whole brain single cell suspensions, it remains to be determined if subtypes of neurons and astrocytes have different LD storage capacities. Moreover, several recent studies suggested LD alterations in the brain with aging or diseased states[11]. The tdTom-Plin2 mouse model will therefore provide a powerful tool to study

age-related LD accumulation. Furthermore, crossing these mice with neurodegenerative mouse models will enable novel approaches to investigate LD changes under pathological conditions, offering insights into the role of LD dynamics in neurodegenerative processes.

LDs are present in NSPCs in vitro[23] and in the SVZ in vivo[15]. Therefore, we focused on the two main adult neurogenic regions, the SVZ and the DG. Within these regions, we show that LDs can be found in neurons, astrocytes, oligodendrocytes, microglia, ependymal cells, endothelial cells and NSPCs. However, not all the cells of a specific cell type contained LDs. It therefore remains to be investigated if this is due to dynamic intracellular LD fluctuations, or intercellular lipid coupling between cells such as that described between neurons and glia in vitro and in *Drosophila*[14,16,17]. Interestingly, almost all NSPCs, as well as astrocytes and neurons in vitro have LDs[23,32], while not all of these cells have LDs in vivo. This discrepancy may be attributed to differences in the microenvironment and nutrient availability between in vitro versus in vivo conditions, or due to varying levels of cellular stress, as LDs can exert protective functions under cellular and oxidative stress[2]. Thus, these differences highlight the importance of considering the metabolic situation in vitro versus in vivo when studying the role of LDs. In addition, we found that LD number and sizes vary a lot among different cell types, with ependymal LDs being much larger than for example LDs in endothelial cells. How this LD heterogeneity influences the different cell types remains to be explored. While PLIN2 is one of the most ubiquitous LD coat proteins, one also has to keep in mind that there might be subpopulations of LDs that do not express *Plin2* and thus will not be visualized by tdTomato in the tdTom-Plin2 mouse.

As previously suggested for astrocytes and microglia[19,52] one reason why not all the cells of a given cell type have LDs in vivo could be that the presence of LDs could be linked to distinct subtypes. Indeed, our scRNA sequencing experiment showed a specific LD-containing subpopulation of microglia under physiological conditions, based on the presence or absence of tdTom-PLIN2, hence presence or absence of LDs detectable by FACS. This population shared some but not all signatures with DAM and LDAM microglia[19,41,53], suggesting that microglia with similar signatures might already start to develop before disease or aging features appear. It will therefore be interesting to see if this LD-containing population changes with aging and becomes more similar to the previously described disease-associated microglia. As a proof-of-concept, we here show that the tdTom-Plin2 mouse allows the unbiased separation of cell populations based on LD content, which can then be further explored by downstream analyses such as scRNA sequencing. Our procedure heavily enriched the microglial population, with more than 90% of the high-quality sequencing reads coming from microglia. This enrichment is most likely due to their resistance to stresses such as mechanical disruption, followed by FACS, and then proceeding with 10x-Genomics single cell droplet generation. To fully exploit this novel model, better protocols including whole cell FACS need to be established to cover more cell types.

LDs can accumulate upon altered lipid availability in several tissues, including the liver[54], and we demonstrate that an increase in LDs can indeed be detected in the liver of tdTom-Plin2 mice after a short-term HFD. Surprisingly, we also found a small, but significant increase in TAGs in the brain of HFD fed mice, indicating that even a short exposure to HFD can have an impact on TAGs in the brain. Several studies have addressed brain lipidome alterations upon diet changes, however, as the HFD exposure was much longer in these studies or the diet composition was different[55,56], it is difficult to directly compare those results. Our findings, showing a slight increase in LDs in the rostral part of the lateral ventricle, suggest that at least certain brain cells can directly react to increased lipid availability with increased lipid storage. This is in line with a study by Ogrodnik and colleagues

who investigated the effects of a 2-4-month long exposure to HFD on the brain of 8-month-old mice. They report an accumulation of PLIN2 in the walls of the lateral ventricles[57], accompanied by a decrease in neurogenesis. However, the precise relationship between these two phenomena requires further exploration. A longer exposure to HFD is needed to address to which extent LDs in different cell types in the brain reacts to changes in circulating lipids.

Relatively little is known about the role of lipid metabolism in brain development. A recent publication showed that de novo lipogenesis, which has been found crucial for adult NSPCs[26], is also critically required in NSPCs during early mouse brain development[58] and human brain development[59]. As de novo lipogenesis directly influences LD accumulation in adult NSPCs[23], we wanted to exploit the tdTom-Plin2 mouse model to address LDs during brain development. We could show that LDs are highly abundant in the developing mouse cortex at E14.5 and can be seen in RGs, in line with scRNA sequencing data of the developing brain[43,60]. Their high abundance might be linked to high de novo lipogenesis, as suggested by the high expression of *Fasn* at this stage of development (http://mousebrain.org[43]). Interestingly, overnight incubation with OA significantly increased LDs, illustrating that external lipids can influence LD abundance. This is in line with a study on cholesterol metabolism during mouse brain development, which showed that LDs increased when cholesterol metabolism was disrupted, potentially due to an increased uptake of lipoproteins from the circulation[61]. Future studies will have to show if there is a regional, temporal and cell type specific variations in LD abundance during brain development and will assess how manipulation of LD build-up or breakdown influences this process. Interestingly, using embryonic mouse brain sections, we demonstrate that the fluorescent LD label allows live imaging of LDs in living tissues ex vivo. While the tracking of individual cells and the detailed changes of LDs within cells over time will require more work, this illustrates the usefulness of this mouse model for studying the dynamics of LDs.

In summary, our findings highlight the tdTom-Plin2 reporter mouse as a valuable novel tool to study LDs and their dynamics. As PLIN2 is ubiquitously expressed, this model is not only interesting for the neuroscience community but can also be used for studying other tissues that contain LDs under physiological or diseased conditions.

## Methods

### Generation of tdTom-Plin2 mice

Single guide RNA (sgRNAs) plasmid: sgRNAs targeting the N-terminus of *Plin2* were designed using https://www.addgene.org/crispr/reference/. Oligos (100 μM, Microsynth) were phosphorylated and annealed in a reaction with Oligos (1:10), T4 Polynucleotide Kinase (1:10, #MO201S, New England Biolabs), T4 Ligase reaction buffer (#B0202S, New England Biolabs) with 10 mM 10x Adenosine 5′-Triphosphate (1:10, #P0756S, New England Biolabs). The reaction was incubated in a thermocycler for 37 °C for 30 min, 95 °C for 5 min and then ramped down to 25 °C, 5 °C at a time.

The sgRNA plasmid backbone, pSpCas9 (BB)−2A-Puro (PX459) v2.0 (Ran et al., 2013), was digested for 4 h at 37 °C using the restriction enzyme BbsI (#R3539S, New England Biolabs) and run on a 1% agarose gel for 3.5 h at 100 V. The digested band was cut out and extracted from the gel using QIAquick Gel Extraction Kit (#28704, Qiagen).

The sgRNA was then ligated into the backbone by incubation of 10 ng backbone, annealed and phosphorylated sgRNA (1:10) and T4 Ligase reaction buffer (1:10, #B0202S, New England Biolabs) with 10 mM 10x Adenosine 5′-Triphosphate (#P0756S, New England Biolabs) for 30 min in RT.

sgRNA oligonucleotide: 5′ CACCGTCCTGGAGAAAAGAGACGCC 3′

tdTomato plasmid: Mouse *Plin2* homology arms were obtained by extracting DNA from mouse primary mouse NSPCs from the SVZ using

DNeasy Blood & Tissue Kit (#69506, Qiagen). The sequences covering the sgRNA target site were extracted from the genomic DNA using PCR, run on a 1% agarose gel and gel extracted using QIAquick Gel Extraction Kit (#28704, Qiagen). *tdTomato* and *neomycin* were cloned out from H3.1-miCOUNT (Denoth-Lippuner et al., 2020) using PCR, run on a 1% agarose gel and gel extracted using QIAquick Gel Extraction Kit (#28704, Qiagen). The plasmid backbone, pFA6 (Janke et al., 2004) was digested using the restriction enzymes SacII (#R0157S, New England Biolabs) and HindIII (#R3104T, New England Biolabs). All fragments were cloned together using the Gibson Assembly® Master Mix (#E2611S, New England Biolabs). Therefore, all primes used for amplifying the fragments were designed to have an overhang to align to the adjacent fragment in the plasmid. The reverse primer for cloning neomycin has a T2A sequence inserted in the overhang linking to *tdTomato*, and the reverse *tdTomato* primer have a GGGGS linker inserted in the overhang linking to *Plin2*. Mutations in the sgRNA target site of the plasmid were introduced and *neomycin* was removed using Q5 Site-Directed Mutagenesis Kit (#E0554S, New England Biolabs).

Once the plasmids were generated, they were sent to PolyGene transgenetics (Switzerland) for the generation of the mouse. In brief, C57Bl/6 ESCs were CRISPRed and grown as colonies. Selected colonies of ESCs were harvested and DNA was extracted. ESCs were screened for the correct insertion of tdTomato using PCR combined with sequencing of the PCR product. The selected correct ESC clone was then screened for the top three most likely off targets (*Grin2b, Sumf1* and *Gpc6*), based on the online prediction tool: https://cm.jefferson.edu/Off-Spotter/, using PCR and sequencing of the PCR product. The most promising clone of ESCs were then selected for C57Bl/6 blastocyst injection to generate chimeras. The chimeras were bred and offspring with germline tdTomato insertion was selected. The tdTom-Plin2 mice were either crossed inter se or crossed with C57Bl/6 mice obtained from Janvier (France).

## Transformation

5-alpha Competent E. coli (High Efficiency) (#C2987I, New England Biolabs) were thawed on ice, 1 μl containing 1 pg- 100 ng DNA was added and incubated on ice for 30 min. The bacteria were then heat shocked at 42 °C for 30 s and then placed back on ice for 5 min. 950 μl SOC outgrowth medium (#B9020S, New England Biolabs) was added and sample was incubated on rotation at 37 °C for 1 h. 100 μl bacterial suspension was plated on agar plates with ampicillin (100 μg/ml) and incubated at 37 °C overnight. Single colonies were picked for further amplification.

## Plasmid amplification

Amplification of plasmids was done using QIAprep Spin Miniprep Kit (#27106, Qiagen) and EndoFree Plasmid Maxi Kit (#12362, Qiagen) according to the manufacturer's instructions.

## Animals

TdTom-Plin2 mice were generated as described above. C57Bl/6 female mice used for timed mating were purchased from Janvier (France). The Nestin-GPF mouse strain is originally described by Yamaguchi and colleagues[42]. The Thy1GFP mouse is originally described by Feng and colleagues[38] and the Aldh1l1GFP mouse by Gong and colleagues[39]. All mice were kept under standard conditions in ventilated cages with ad libitum food and water. Housing conditions were as follows: dark/light cycle 12/12, ambient temperature around 21–22 °C, and humidity between 40 and 70% (55% on average).

Mice fed a HFD (60% energy from fat, SAFE® Complete Care Competence, irradiated #292HF) were housed in the standard conditions, but fed a HFD for 4 weeks starting at the age of 8 weeks. Body weight was monitored on weekly basis, EchoMRI™ was performed at the start and endpoint of the HDF fed mice and the control group fed standard chow diet (#SAFE-150, SAFE® Complete Care Competence).

All experiments including animals were carried out in compliance with the Swiss law after approval from the local authorities (Cantonal veterinary office, Canton de Vaud, Switzerland).

## Collection of mouse tissue

*Perfusion:* Mice were anesthetized by i.p. injection of pentobarbital (150 mg/kg) and intracardially perfused with ice cold 0.9% saline until no blood remained, then perfused with ~ 30 ml ice cold 4% PFA, followed by another ~10 ml 0.9% saline. Brain and body were collected and post fixed in ice cold 4% PFA overnight. Mice exposed to HFD were only perfused with 40 ml 0.9% ice cold saline, one brain hemisphere and part of the liver was post fixed in 4% PFA overnight, and the other hemisphere and part of the liver were snap frozen. Serum was collected from mice exposed to HFD by aspiring blood from the ventricle of the heart just before the saline perfusion was started. The blood was transferred to K-EDTA tubes, inverted 5 times and stored on ice for 5-20 min before centrifugation at 1000 x *g* for 10 min (4 °C). The serum fraction was gently collected with a pipette transferred to a fresh Eppendorf tube and snap frozen.

*Subventricular zone whole mount:* The subventricular zone whole mount was performed as previously described[62,63]. In brief, mice were sacrificed by decapitation and the lateral walls of the lateral ventricle were dissected out and post fixed in 4% PFA overnight. The ventricle walls were washed 3 times in 0.1 M PBS before immunohistochemical staining.

*Sample collection and preparation for lipidomics, RT-qPCR and Western blot:* The liver and one snap frozen brain hemisphere (without olfactory bulb) were homogenized using Evolution & Cryolys® Evolution (#P000062-PEVO0-A.0 & #P000671-CLYS2-A.0, Precellys®), Stainless Metaltube (#P000952-LYSK0-A.0, Precellys®) and 6.8 mm ceramic bead (#P000931-LYSK0-A.0, Precellys®). All sample handling was done in liquid nitrogen. The frozen tissue was placed in a metal tube together with the ceramic bead. The tube was placed in the cryogrinding machine and homogenized at 8800 rpm for 5 sec. The homogenized brain powder was aliquoted for lipidomic analysis, RT-qPCR analysis and western blot. The liver powder was aliquoted for RT-qPCR and western blot.

For the expression analysis of Plin2 in different organs, 3 tdTom-Plin2 het male mice were intracardially perfused with ice cold 0.9% saline until no blood remained. Organs were dissected and immediately frozen on dry ice. Tissues were mechanically disrupted by passing them through syringes with decreasing needle size (G20 and G25) in RLT lysis buffer supplemented with beta-mercaptoethanol, and RNA extraction was performed with the RNeasy mini kit (#74134, Qiagen) according to the manufacturer's instructions.

## Histological evaluation

Mice were anesthetized by i.p. injection of pentobarbital (150 mg/kg) and intracardially perfused with ice cold 0.9% saline until no blood remained, then perfused with ~ 30 ml ice cold PFA, 4% followed by another ~10 ml 0.9% saline. The thoracic and abdominal cavities were opened, and further fixation was achieved by placing the mice in a 10% formalin solution. Each mouse was subjected to a macroscopic pathological examination and extensive sampling for histological evaluation. These samples were routine paraffin wax embedded and sections were prepared and stained with hematoxylin and eosin (H&E). In addition, a sample of liver from each mouse was frozen, and an OCT block was prepared. From this block, two sections of liver, separated by 10-20 μm, were prepared and stained with Oil Red O (ORO). The H&E and ORO stained slides were reviewed by a pathologist who was blinded to the groups.

## Brain cell extraction

*Neural stem/progenitor cell extraction:* Adult mouse NSPCs from the SVZ of three 8-week-old female mice were isolated using papain-based

MACS Neural Tissue Dissociation Kit (#130-092-628, Milteny) and the GentleMacs Dissociator (Milteny) according to the manufacturer's instructions.

In brief, mice were shortly anaesthetized with isoflurane, followed by decapitation. SVZ were micro-dissected, and a single cell suspension was generated using the papain-based MACS Neural Tissue Dissociation Kit (#130-092-628, Milteny), according to the manufacturer's instructions. The single-cell suspension was followed by myelin removal using the MACs myelin removal beads (#130-096-731, Milteny) and a QuadroMACS Separator (#130-090-976, Milteny) according to the manufacturer's instructions. The obtained cells were cultured as neurospheres in DMEM/F12/GlutaMAX (#31331-028, Gibco) with B27 (#17504044, Gibco), 20 ng/ml human EGF (#AF-100-15, PeproTech), 20 ng/ml human basic FGF-2 (#100-18B, PeproTech), and 1x PSF (#15240062, Gibco). The medium was changed every 2-3 days. The neurospheres were expanded for 5 passages to remove progenitors and other proliferating cells. After 5 passages, cells were changed to the following culture medium: DMEM/F12/GlutaMAX (#31331-028, Gibco), N2 (#11520536, Gibco), 20 ng/ml human EGF (#AF-100-15, PeproTech), 20 ng/ml human basic FGF-2 (#100-18B, PeproTech), 5 mg/ml Heparin (#H3149-50KU, Sigma) and 1x PSF (#15240062, Gibco). All the in vitro experiments were done on pooled NSPCs from 3 mice, passage 7-15.

Whole brain cell extraction: Protocol adapted from ref. 64. In brief, mice were anesthetized by i.p. injection of pentobarbital (150 mg/kg) and intracardially perfused with 30 ml cold PBS (pH 7.3–7.4). The brain was collected and finely chopped with a few drops of Hibernate A (Hibernate®A without phenol red, BrainBits) before complete dissociation in 1.5 ml Hibernate A in a 1 ml douncer (loose pestle). The cell suspension was filtered twice in a prewet 70 µm cell strainer (#15370801, Fisher Scientific), the rubber tip of a 1 ml syringe was used to facilitate the filtering. The cell suspension was transferred to a precooled 15 ml tube and centrifuged at 500 x $g$ for 6 min. The supernatant was aspirated, and the cell pellet was resuspended in 2 ml cold PBS (pH 7.3–7.4), before adding 1 ml isotonic percoll in PBS (pH 7.3–7.4) (#GE17-0891-01, Sigma-Aldrich). The suspension was mixed well before 4 ml cold PBS (pH 7.3–7.4) was gently added on top of the cell suspension. The layered sample was centrifuged at 3000 x $g$, 10 min. The supernatant and the myelin disk were gently aspirated. The cell pellet was gently washed by adding 4 ml cold PBS (pH 7.3–7.4) and gently tilting the tube back and forth before centrifuging at 450 x $g$, 10 min. Cell pellets used for FACS were dissolved in 500 ul EDTA-DPBS (#E8008, Merck Millipore), cell pellets used for scRNA sequencing were dissolved in 500 µl PBS (pH 7.3–7.4).

ScRNA seq and analysis: After sorting, cells were immediately processed according to the 10X Chromium protocol. Briefly, an appropriate volume of each cell suspension containing cells from tdTomato positive and tdTomato negative conditions were combined with 10X Chromium reagent mix and samples were loaded into two separate lanes. Cell capture, lysis, mRNA reverse transcription, cDNA amplification and libraries were performed following 10X Genomics Chromium dual indexing Single Cell 3' V3.1 reagent kit instructions. Libraries were then multiplexed and sequenced according manufacture recommendations with paired-end reads using a HiSeq4000 platform (Illumina) with an expected depth of 100'000 reads per single cells. All the sequencing experiments were performed within the Genomics Core Facility of the University of Lausanne. Alignment of sequenced reads to the mouse genome (GRCm38) and filtered gene–barcode matrices were realized by running Cell Ranger Single-Cell Software Suite v5.0.1 (10X Genomics). The cell ranger count function was used to generate filtered gene/cell expression UMI corrected matrices by selecting probable cells and removing empty lipid droplets. To filter only high-quality cells, we applied selection based on mitochondrial genes percentage (<10%) and number of genes per cell (>500 genes), doublets cells were removed using Scrublet package. After applying these filters, 845 cells for "tdTomato positive" condition and 784 cells for "tdTomato negative" condition were kept for further analysis. The cells were then annotated using reference single cell atlas, and microglial cells were kept for following analysis (812 microglia for "tdTomato positive" condition and 766 microglia for "tdTomato negative" condition). The raw count matrix was normalized and scale using SCTransform procedure from Seurat package. For UMAP visualization, dimensionality reduction was performed using standard function from Seurat. We first adopted a graph-based clustering approach using "FindClusters" function with a resolution of 0.3. Differentially expressed genes were identified based on their weight, resulting from differential pairwise expression analysis using Seurat "FindAllMarkers" function with default parameters. The identified gene candidates for each condition were interrogated for statistically significant gene ontologies analysis using the free online tool Metascape (metascape.org).

### Cell culture

NSPCs were grown on uncoated plastic cell culture dishes for expansion (#430167, Corning, 100 mm * 20 mm, TC-treated). Cells used for experiments were plated on glass coverslips (#10337423, Fisher) or standard plastic cell culture dishes coated with poly-L-ornithine (#P3655, Sigma) and laminin (#L2020-1MG, Sigma). Proliferating NSPCs were kept in DMEM/F12/GlutaMAX (#31331-028, Gibco) complemented with N2 (#11520536, Gibco), 20 ng/ml human EGF (#AF-100-15, PeproTech), 20 ng/ml human basic FGF-2 (#100-18B, PeproTech), 5 mg/ml Heparin (#H3149-50KU, Sigma) and 1x PSF (#15240062, Gibco). Medium was changed every 2-3 days. EdU pulse was done by incubating the cells with EdU (1:1000) (2.52 mg/ml, Click-iT Plus EdU Alexa Fluor 647 Imaging Kit, # 15224959, Invitrogen) for 1 h at 37 °C before fixation with 4% PFA.

### Metabolic measurements

Oxygen consumption rate (OCR) and extracellular acidification rate (ECAR) of NSPCs was measured using the Seahorse XF Cell Energy Phenotype Test Kit (Agilent, # 103325-100), according to the manufacturer's instructions. Measurements were performed on a Seahorse XFe96 analyzer (Agilent) in a 96-well plate format. In brief, 40'000 cells were plated on coated Seahorse 96-well plates and left to attach overnight, in 80 µl proliferation medium per well. Cells were then washed 2 times with XF DMEM assay medium (Agilent, #103575-100), supplemented with glucose (Agilent, 103577-100 Seahorse XF 1.0 M glucose solution), glutamine (Agilent, 103579-100 Seahorse XF 200 mM glutamine solution) and pyruvate (Agilent, 103578-100 Seahorse XF 100 mM pyruvate solution) with final concentrations as following: glucose 10 mM, glutamine 2 mM, pyruvat 1 mM. Before starting the measurements, NSPCs where incubated for 1 h in a non-C02 but humidified 37 °C incubator (Agilent) in the same XF assay medium as described above, including 20 ng/ml human EGF and 20 ng/ml human basic FGF-2. Baseline OCR and ECAR were measured over 3 timepoints (15 min). Oligomycin (1 µM) and FCCP (1 µM) were injected into each well using the Seahorse Cartridge system and OCR and ECAR measured immediately after.

At the end of the measurements, live Hoechst 33342 was injected in each well and cells were incubated for 15 min to stain the nuclei. Cells were fixed with 2% PFA and images of the entire wells were taken with a Thunder microscope, 5x objective. The area covered by the Hoechst signal was calculated using ImageJ, and this value was used for the normalization of the OCR and ECAR values of each well. For the baseline, the 3rd measure (after 14.5 min) was used. For the metabolic capacity, the difference between the first measure after the oligomycin and FCCP injection and the baseline was calculated. A total of 12–15 replicates per condition were averaged, and the experiments were repeated 3 times.

## Proteomics analyses

Sample preparation and protein digestion: NSPCs from tdTom-Plin2 het and wt littermates were cultured in parallel as described under cell culture. Cell pellets of 4 consecutive passages were collected, washed 2x with PBS and snap frozen on dry ice. Samples were digested following a modified version of the iST method[65](named miST method). Washed cell pellets were lysed in 300ul miST lysis buffer (1% Sodium deoxycholate, 100 mM Tris pH 8.6, 10 mM DTT), heated for 10 min at 75 C and disrupted by tip sonication (2x15s). Samples were heated 5 min at 95 °C, diluted 1:1 (v:v) with water containing 4 mM MgCl2 and benzonase (Merck #70746, 100x dil of stock = 250 Units/ul), and incubated for 15 minutes at RT to digest nucleic acids. Based on tryptophane fluorescence quantification[66], 50 ug of proteins at 1 ug/ul, were used for digestion. Reduced disulfides were alkylated by adding ¼ vol. of 160 mM chloroacetamide (32 mM final) and incubating for 45 min at RT in the dark. Samples were adjusted to 3 mM EDTA and digested with 0.5 ug Trypsin/LysC mix (Promega #V5073) for 1 h at 37 °C, followed by a second 2 h digestion with an additional 0.5ug of proteases. To remove sodium deoxycholate, two sample volumes of isopropanol containing 1% TFA were added to the digests, and the samples were desalted on a strong cation exchange (SCX) plate (Oasis MCX; Waters Corp., Milford, MA) by centrifugation. After washing with isopropanol/1%TFA, peptides were eluted in 200ul of 80% MeCN, 19% water, 1% (v/v) ammonia, and dried by centrifugal evaporation.

Mass Spectrometry analysis: LC-MS/MS analyses were carried out on a TIMS-TOF Pro (Bruker, Bremen, Germany) mass spectrometer interfaced through a nanospray ion source ("captive spray") to an Ultimate 3000 RSLCnano HPLC system (Dionex). Peptides (200 ng) were separated on a reversed-phase custom packed 45 cm C18 column (75 µm ID, 100 Å, Reprosil Pur 1.9 um particles, Dr. Maisch, Germany) at a flow rate of 0.250 ul/min with a 2-27% acetonitrile gradient in 93 min followed by a ramp to 45% in 15 min and to 90% in 5 min (total method time: 140 min, all solvents contained 0.1% formic acid). Identical LC gradients were used for DDA and DIA measurements. For the creation of the spectral library, data-dependent acquisitions (DDA) were carried out on the 6 bRP fractions sample pool using a standard TIMS PASEF method[67] with ion accumulation for 100 ms for each survey MS1 scan and the TIMS-coupled MS2 scans. The duty cycle was kept at 100%. Up to 10 precursors were targeted per TIMS scan. Precursor isolation was done with 2 Th or 3 Th windows below or above m/z 800, respectively. The minimum threshold intensity for precursor selection was 2500. If the inclusion list allowed it, precursors were targeted more than one time to reach a minimum target total intensity of 20'000. The collision energy was ramped linearly based uniquely on the 1/k0 values from 20 (at 1/k0 = 0.6) to 59 eV (at 1/k0 = 1.6). The total duration of a scan cycle, including one survey and 10 MS2 TIMS scans, was 1.16 s. Precursors could be targeted again in subsequent cycles if their signal increased by a factor of 4.0 or more. After selection in one cycle, precursors were excluded from further selection for 60 s. Mass resolution in all MS measurements was approximately 35'000.

The data-independent acquisition (DIA) used mostly the same instrument parameters as the DDA method and was as reported previously[68]. Per cycle, the mass range 400-1200 m/z was covered by a total of 32 windows, each 25 Th wide and a 1/k0 range of 0.3. Collision energy and resolution settings were the same as in the DDA method. Two windows were acquired per TIMS scan (100 ms) so that the total cycle time was 1.7 s.

Data processing: Raw Bruker MS data were processed directly with Spectronaut 18.1.230626.50606 (Biognosys, Schlieren, Switzerland). A library was constructed from the DDA bRP fraction data by searching the reference annotated mouse proteome database of July 1st, 2022 (uniprot_sprot_2022-01-07_MOUSE.fasta, www.uniprot.org). For identification, peptides of 7-52 AA length were considered, cleaved with trypsin/P specificity and a maximum of 2 missed cleavages.

Carbamidomethylation of cysteine (fixed), methionine oxidation and N-terminal protein acetylation (variable) were the modifications applied. Mass calibration was dynamic and based on a first database search. The Pulsar engine was used for peptide identification. Protein inference was performed with the IDPicker algorithm. Spectra, peptide and protein identifications were all filtered at 1% FDR against a decoy database. Specific filtering for library construction removed fragments corresponding to less than 3 AA and fragments outside the 300-1800 m/z range. Also, only fragments with a minimum base peak intensity of 5% were kept. Precursors with less than 3 fragments were also eliminated and only the best 6 fragments were kept per precursor. No filtering was done on the basis of charge state and a maximum of 2 missed cleavages was allowed. Shared (non proteotypic) peptides were kept. The resulting library included 154'856 precursors (109'255 proteotypic peptides) mapped to 8'558 protein groups. Peptide-centric analysis of DIA data was done with Spectronaut 18.1 using the library described above. Single hits proteins (defined as matched by one stripped sequence only) were kept in the Spectronaut analysis. Peptide quantitation was based on XIC area, for which a minimum of 1 and a maximum of 3 (the 3 best) precursors were considered for each peptide, from which the median value was selected. Quantities for protein groups were derived from inter-run peptide ratios based on MaxLFQ algorithm[69]. Global normalization of runs/samples was done based on the median of peptides. The raw output table contained 8'243 protein groups (minimum 1 precursor per group).

All subsequent analyses were done with the Perseus software package (version 1.6.15.0)[70]. Contaminant proteins were removed, and quantity values were log2-transformed. After assignment to groups, only proteins quantified in at least 4 samples of one group (8118 protein groups). After missing values imputation (based on normal distribution using Perseus default parameters), t-tests were carried out among all conditions, with Benjamini-Hochberg correction for multiple testing (Q-value threshold <0.05). For differential analysis, proteins with a minimum of 2 peptides detected were kept (7641) and proteins with a twofold change and an adjusted $p$-value < 0.05 were considered significantly altered.

## Lipidomic analysis

Lipidomic analysis of brains (analysing around 1200 lipid species belonging to five different subclasses including TAGs, DAGs, MAGs, CEs, sphingolipids, glycerophospholipids, free fatty acids) was done as previously described[71]. In brief, lipids were extracted from 25 mg brain tissue sample prepared as described above. The extracted lipids were then analysed using Hydrophilic Interaction Liquid Chromatography coupled to tandem mass spectrometry (HILIC · MS/MS) in positive ionization mode using an TSQ Altis triple quadrupole instrument (Thermo Fisher Scientific) followed by chromatographic separation carried out on an Acquity BEH Amide, 1.7 µm, 100 mm × 2.1 mm I.D. column (Waters). Raw LC-MS/MS data was processed using the MultiQuant Software (version 3.0.3, Sciex technologies). Relative quantification of metabolites was based on Extracted Ion Chromatogram areas for the monitored MRM transitions. The obtained tables (peak areas of detected metabolites across all samples) were exported to "R" software (http://cran.r-project.org/) where the signal intensity drift correction was done within the LOWESS/Spline normalization program followed by noise filtering (CV (QC features) > 30%) and visual inspection of linear response. Lipid concentrations were normalized to sample protein concentration.

## FACS

Isolation of NSPC from the SVZ and hippocampi from 3 male NesGFP x tdTom-Plin2 mice, one NesGFP mouse and one tdTom-Plin2 mouse was performed as described above. But instead of plating cells for propagation, they were dissolved in EDTA-DPBS (#E8008, Merck Millipore), and stained with the viability markers DAPI (1 µg/ml) and

RedDOT (1:500, #40060-T, Biotium) before being sorted on a FAC-SAria II (BD Biosciences).

Whole brain cell isolation from 3 male Aldh1l1GFP x tdTom-Plin2 mice, 3 male Thy1GFP x tdTom-Plin2 mice, one male Aldh1l1GFP mouse, one male Thy1GFP mouse and 2 male tdTom-Plin2 mice were sacrificed by decapitation and performed as described above but without anesthetization and perfusion. Cell pellets were dissolved in EDTA-DPBS and stained with DAPI and RedDOT before being sorted on a FACSAria II.

## Immunocytochemical and immunohistochemical staining

Cells were fixed with 4% PFA (37 °C) for 15-30 min at RT, washed 2 times with PBS and stored at 4 °C. Antibodies and their dilution factor are as following: Actin mouse (1:5000, Sigma-Aldrich A2228), beta-catenin rabbit (1:200, Cell signaling #9587 S), beta-catenin mouse (1:200 BD 610154), CD31 rat (1:100 BD 550274), DCX guinea pig (1:500 Merck Millipore AB253), EGFR goat (1:100 R&D BAF1280), EGFR rabbit (1:100 Abcam ab52894), GFAP chicken (1:600 Millipore AB5541), GS rabbit (1:100 Abcam ab73593), IBA1 rabbit (1:1000, Wako Chemicals 019-19741), NG2 rabbit (1:100 Millipore ab5320), OLIG2 rabbit (1:100, Millipore AB9610), PLIN2 rabbit (1:1000 Abcam ab52356), S100b mouse (1:100, Sigma-Aldrich S2532), SOX2 goat (1:500, R&D Systems AF2018-SP), tdTomato goat (1:500, Sicgen AB8181-200).

Immunocytochemical staining were performed as described[44]. In brief for saponin protocol, cells were incubated in blocking buffer (1.5% Glycine, 3% BSA, 0.01% Saponin in PBS) for 45 min in RT. Followed by incubation in primary antibody diluted in antibody diluent (0.1% BSA, 0.01% Saponin in PBS) overnight, 4 °C. After 3 washes in PBS, cells were incubated in secondary antibody diluted in antibody diluent for at least 1 h in RT, protected from light. Cells were washed 2 times with PBS. Thereafter incubated with DAPI (D9542, Sigma) diluted in TBS for 10 min, followed by one wash in TBS. Coverslips were mounted with a homemade PVA-DABCO-based mounting medium. EdU staining was done using the Click-iT Plus EdU Alexa Fluor 647 Imaging Kit (#15224959, Invitrogen) according to the manufacturer's instructions.

Post fixed brains were, if cut on a microtome, incubated in 30% sucrose overnight before being cut into 40 μm sagittal sections on a microtome. Coronal sectioning was done on a vibratome, generating 25 μm sections. Mice for HFD experiment were embedded in 4% agar before coronal sectioning. Brains from E14.5 embryos were incubated in 30% sucrose and cut in 60 μm coronal sections on a microtome. The sections were then stored in cryopreservation solution (25% ethylene glycol, 25% glycerol in 0.05 M phosphate buffer) at 4 °C.

Immunohistochemical staining was done as follows. Brain sections were washed two times 5 min in PBS on a shaker before incubation in blocking buffer (0.05-0.25% Triton X-100, 3% donkey serum in TBS) for 1 h on a shaker, RT. Sections were then incubated in primary antibody in blocking buffer at 4 °C overnight on a shaker. The brain sections were washed 3 times 10 min with PBS and then incubated in secondary antibody in blocking buffer for at least 1 h on shaker in RT. The sections were then washed one time 10 min in PBS and one time 10 min in TBS before incubation with DAPI in TBS for 10 min. Brain sections were then mounted on Superfrost (#10149870, Fisher Scientific) and homemade PVA-DABCO-based mounting medium.

When detecting LDs, staining was performed according to the previously mentioned procedure, but blocking was done using the saponin blocking buffer (1.5% Glycine, 3% BSA, 0.01% Saponin in PBS) and antibodies were incubated in antibody diluent (0.1% BSA, 0.01% Saponin in PBS).

Livers were incubated in 30% sucrose for 24 h before being cut into 40 μm sections on a microtome. Staining was performed on free-floating sections using the saponin protocol. BD493 (#11540326, Fisher Scientific, in DMSO) was added 1:1000 together with DAPI in antibody diluent and sections were incubated for 2 h at RT on an orbital shaker.

Sections were washed 3 times 10 min in PBS and sections were mounted as described above. Livers from 1 tdTom-Plin2 SD and 1 Ctrl HFD were excluded due to bad tissue quality after cutting. To assess the endogenous tdTomato signal, sections were incubated for 10 min in the blocking buffer from the saponin protocol, nuclei were stained with DAPI followed by 2 times 10 min wash in TBS and sections were mounted as described above.

## RT-qPCR

NSPC pellets from cultured cells were snap frozen on dry ice. RNA was extracted using RNeasy® plus mini kit (#74134, Qiagen) according to the manufacturer's instructions. cDNA preparation was done using the SuperScript™ IV First-Strand Synthesis System with oligo-dT primers (#15327696, Invitrogen). RT-qPCR was performed using TaqMan™ Fast Advanced Master Mix (#4444557, Thermo Fischer Scientific) and Applied Biosystems TaqMan Assays. The following probes were used: Plin2 (Mm00475794_m1), tdTomato (Mr07319439), 18 S (4333760 T), b-Actin (Mm01205647_g), Atgl (Mm00503040), Hsl (Mm00495359_m1), Mgl Mm00449274_m1. Results were analysed using the ddCT method, normalizing the samples to either eukaryotic 18 S rRNA or mouse *beta-actin*. Statistical analyses were performed on the dCT values.

## Western Blot

NSPCs were washed once with PBS, collected, and spun down at 300 x *g* for 3 min. The supernatant was aspired and the cell pellet was resuspended in RIPA buffer (#R0278-50ML, Sigma-Aldrich) with protease inhibitors (#11873580001, Sigma) followed by incubation at 4 °C for 1 h. The cell lysate was then centrifuged at 12000 x g for 15 min at 4 °C. The supernatant was collected, and the protein concentration was measured using Pierce BCA Protein Assay Kit (#23227, Fisher Scientific) according to the manufacturer's instruction.

Medium right lobes of the liver were snap-frozen in liquid nitrogen directly after perfusion with 0.9% saline. All mice were sacrificed at the same time of the day, exactly 4 weeks after starting the diet. For protein extraction, medium lobes of liver were first mechanically lysed with a pestle followed by a G18 needle in lysis buffer. The lysis buffer consisted of RIPA (Sigma, R0278-50ML) and protease inhibitors (Sigma, 11873580001) and was used at a ratio of 500ul per 30 mg of tissue. The continuation of the lysis was carried out at 4 °C on a shaker for 1 hour. After centrifugation at 4 °C at 12000 g for 15 minutes the supernatant was transferred, and protein concentration was measured using the Pierce BCA Protein Assay Kit (#23227, Fisher Scientific). The protein samples were mixed with Laemmli sample buffer (#1610747, Bio-rad) supplemented with beta-mercaptoethanol and denatured at 95 °C for 5 min before loading on a 12% PAA precast gel (#4561044, Bio-Rad). The gel was run at 70 V for 15 min and then 120 V. The proteins were then transferred to a Amersham™ Protran® Western-Blotting-Membrane (#GE10600044, Sigma-Aldrich) at 120 V for 1 h. The membrane was then incubated in a blocking solution (5% milk powder, 0.05% Triton in TBS) followed by overnight incubation of primary antibody in the blocking solution. The membrane was washed 3 times, 10 min, in TBS before incubation of secondary antibody in blocking buffer. The membrane was washed 3 times for 10 min in TBS before being revealed using ECL technology with WesternBright ECL HRP Substrate (K-12045-D20, Witec) or using a LI-COR Odyssey Imaging System. The following antibodies were used: anti-Perilipin 2 rabbit antibody (1:1000, Abcam ab52356), anti-Perilipin 3 goat antibody (1:1000, Abcam ab118605), anti-RFP rabbit antibody (1:1000, Abcam ab62341) anti-β-Actin mouse antibody (1:5000, Sigma-Aldrich A2228), peroxidase AffiniPure donkey anti-rabbit IgG (1:5000, Jackson ImmunoResearch 711-035-152), goat anti-mouse IgG cross-adsorbed secondary antibody DyLight 800 (1:5000, Invitrogen SA5-10176).

## Serum analysis

Serum analysis of ALAT and ASAT, Glucose, Total Cholesterol, High-Density Lipoprotein, Low-Density Lipoprotein, Triglycerides, and Free Fatty Acids was done with the Dimension Xpand Plus automated chemistry system (Siemens Healthcare Diagnostics AG) at Center of PhenoGenomics, EPFL (Switzerland).

## Live tissue preparation and live imaging of LDs in E14.5 brain sections

Time mated TdTom-Plin2 females were sacrificed at day 14.5 after mating through anaesthetisation with isoflurane followed by decapitation. The uterine horns were exposed by caesarean cut and embryo heads were collected in ice-cold $Ca^{2+}/Mg^{2+}$-free Hank's balanced salt solution (HBSS, Gibco). Brains were rapidly dissected out and embedded in 3% low-melting point agarose (LMP-agarose, #6351.2, Roth). Coronal 250 μm-thick slices at the level of the somatosensory cortex were then cut in ice-cold HBSS on a Vibratome (#VT1000S, Leica) and kept in their surrounding agarose. Slices were then placed in an incubator (37 °C, 5% $CO_2$) for at least one hour of recovery on floating nucleopore track-etched polycarbonate membranes (∅13 mm, 1 μm pore size, #WHA110410, Whatman) in supplemented DMEM-F12 medium (2% B27, 2 mM GlutaMAX, 1% penicillin-streptomycin and 1% N2). Prior to imaging, slices were labelled with SPY650-DNA (1:1000, #SC501, Spirochrome) for 1 h and CellTrace™ CFSE (1:1000, #C34554, ThermoFisher) for 30 min. Slices were then washed, transferred, anchored and superfused with warm (37 °C) and bubbled (medical oxycarbon) aCSF (125 mM NaCl, 26 mM $NaHCO_3$, 10 mM D-Glucose, 3 mM KCl, 1.5 mM $MgCl_2$, 1.25 mM $NaH_2PO_4$, 1.6 mM $CaCl_2$) in a standard cover glass-mounted chamber (#64-0265, Warner Instruments) on an inverted confocal microscope (Nikon A1r), equipped with oil-immersion 60x objective (1.4 Plan Apo VC H, Nikon). 30 μm-thick stacks (2 μm-stepped) were acquired with a 2x numerical zoom every 5 min with resonant laser scanning in the VZ of the dorsal pallium. For OA experiments, slices were incubated in 0.1 μM OA (coupled to fatty acid free BSA).

## Imaging and image analysis

All images used for LD quantification were acquired with either a Leica Thunder Imaging System, a confocal microscope (Zeiss, 780, 800, and 900), or a spinning disk confocal (Nikon Ti2, Yokogawa CSU-W1) with a 10x, 40x, or 63x objective with digital zoom. Images for assessing cell proliferation in vitro was acquired using an epifluorescent microscope (Nikon, 90i) with a 20x objective.

In vitro colocalization of tdTomato and PLIN2 was assessed using Fiji (Version 2.0.0-rc-69/1.52p). In brief, the images were pre-processed using "Z-project", "8-bit conversion", "Subtract Background", "Brightness/Contrast", "Set threshold" and converted to mask. Both tdTomato and PLIN2 was then quantified using "Analyse Particles", and a mask was created. Colocalization of the masks of the two channels were assessed using "colocalization". The number of EdU+ cells were also assessed using Fiji. The EdU image was processed as follows "8-bit conversion", "smooth", "enhance contrast", "set threshold", conversation to mask, "fill holes" and "watershed". The mask was then quantified using "Analyse Particles". PLIN2 in vitro and in vivo was quantified by "Analyse Particles", after the same pre-processing as for assessing colocalization of tdTomato and PLIN2. The number of cells in all conditions was quantified through manual counting of DAPI positive nuclei.

TdTomato intensity (enhances with staining against RFP) along the VZ was measured using Fiji (Version 2.0.0-rc-69/1.52p). In brief, images were rotated, so that the VZ (Sox2 + ) lays horizontal. Images were processed "8-bit conversion", "subtract background 5 px". On the composite image (Sox2+ and tdTom + ), 1 or several lines with 200 px width and 100 μm length (linear region of interest, ROI) were drawn perpendicular to the VZ and the "plot profile" function was used to

obtain the average gray values over distance to VZ. Values were averaged per embryo and normalized to the first value to obtain % changes.

Area covered of tdTom-PLIN2 in wholemount preparations of the lateral ventricles were quantified using Fiji (Version 2.0.0-rc-69/1.52p). In brief, the images were processed using "Color Balance" to remove the beta-Catenin signal, "8-bit conversion", "Brightness/Contrast", "Set threshold" and converted to mask. The mask was then divided into 8 zones in which the tdTom-Plin2 signal was quantified using "Measure".

Area covered of BD493 in liver sections was quantified using Fiji (Version 2.0.0-rc-69/1.52p). In brief, the images were processed using "8-bit conversion", "Set threshold" and converted to mask. BD493 was then quantified using "Analyse Particles" and normalized to the DAPI area. The area covered of DAPI was processed using "8-bit conversion", "Gaussian Blur", "Set threshold" and converted to mask before being quantified using "Measure". 1 tdTom-Plin2 SD mouse was excluded from the liver analysis after the Grubbs test in Prism (GraphPad) identified it as a significant outlier ($p < 0.05$).

Area covered of tdTom-PLIN2 in liver sections was quantified using Fiji (Version 2.0.0-rc-69/1.52p). In brief, the images were processed using "Brightness/Contrast", "Subtract Background" and then the mean intensity was quantified using "Measure" and normalized to the DAPI area.

All quantifications were done blinded.

## Statistics

Statistical analyses were performed with Prism (GraphPad) as following. A student t-test was used for comparing two groups. When comparing more than two groups analyses were done using a 2-way ANOVA followed by posthoc tests (Sidaks multiple comparison test). Significance was considered for $p$-values < 0.05. For fold change (FC) analyses compared to a control group, FC values were log2 transformed and a one-sample t-test was performed. The nature of the sampling ("n") is described in each figure legends. A minimum of $n = 3$ is used for each statistical comparison. All $p$-values, test descriptions and raw data are provided in the Source Data file.

## Reporting summary

Further information on research design is available in the Nature Portfolio Reporting Summary linked to this article.

## Data availability

The data supporting the findings from this study are available within the article file and its supplementary information. scRNA-seq raw and preprocessed data generated in this study have been deposited in the GEO database under accession code GSE267069. All raw MS data together with raw output tables are available via the Proteomexchange data repository (www.proteomexchange.org) with the accession PXD051943. Any other raw data or noncommercial material used in this study are available from the corresponding author upon request. Source data are provided with this paper.

## Code availability

Details of the software codes used are described in the method section, provided with the raw data GSE267069 and are available from the corresponding author on request.

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

## Acknowledgements

We would like to thank A. Denoth Lippuner and Sebastian Jessberger for kindly providing plasmids and input used for the CRISPR approach, R. C. Paolicelli for providing the Thy1GFP mouse and helpful input on the microglia dataset, and P. Bezzi for providing the Aldh1l1GFP mouse. We furthermore thank D. Tavel, M. Niquille and V. Scandella for technical help. We would like to thank the Flow Cytometry facility at University of Lausanne, especially D. Labes for technical help and help with analysis, the Metabolomics Unit at University of Lausanne, specifically T. Taev for technical help and H. Gallart-Ayala for help with analysis. We also thank the Cellular Imaging facility at University of Lausanne for technical help. Mass spectrometry-based proteomics work was performed by the Protein Analysis Facility at the University of Lausanne, and we specifically thank M. Quadroni for the analysis and helpful input. We further thank the Center of PhenoGenomics, EPFL, specifically R. Combe for technical help. We are also grateful to the laboratory team in the Histology Laboratory, Institute of Veterinary Pathology, Vetsuisse Faculty, University of Zurich, for technical support. This work was supported by funding from the University of Lausanne and the Swiss National Science Foundation (grant # 31003A_175570, M.K), University of Basel and European Research Council Advanced Grant (No 789328, F.D).

## Author contributions

S.M. designed, performed, analysed and interpreted experiments. A.C.D. performed SVZ whole mount experiments and characterized the SVZ. C.C. designed and performed embryonic live imaging experiments, together with S.M. and C.M.I. ScRNA seq experiments were performed and analysed by S.M., E.M., M.K. and L T. V.M., performed and analyzed metabolic measurements, helped with in vitro and in vivo work, and performed liver analyses. F. B. performed liver experiments and analysed whole mount data. S.D.N. performed the histopathological analyses. D. J. and F. D. provided resources and interpreted data. M.K. developed the concept and designed, performed, analysed, and interpreted experiments. S. M. and M. K. wrote the manuscript, with input from all authors.

## Competing interests

The authors declare no competing interests.
