## [Peer Review File · Nature Communications]

A fluorescent perilipin 2 knock-in mouse model reveals a high abundance of lipid droplets in the developing and adult brainEditorial Note: Parts of this Peer Review File have been redacted as indicated to remove third-party material where no permission to publish could be obtained.

Reviewer #1 (Remarks to the Author):

Lipid droplets are important lipid storage organelles that play a critical role in cell physiology and consequently many diseases. Here, Madsen et al. developed a transgenic mouse line to label lipid droplets. They endogenously tagged the lipid droplet resident protein PLIN2 with tdTomato. They show that their tagging strategy does not disrupt PLIN2 function as they observe no major effects on lipid metabolism. They demonstrate the usefulness of their model system in different organ systems (liver and the brain) and in fixed tissue, cultured cells, and ex vivo brain slices.

Overall, I think there is great value in developing this mouse line. It will be useful for biologists across different physiological systems. Particularly in the brain, where the role of lipid droplets is a rapidly growing field. I think the study is well written and for the most part, well-controlled. However, there are a few technical issues that need to be addressed first. Notably, Z-stacks need to be performed to conclude the cell-types that contain the lipid droplets. And the authors need to better reconcile the sheer abundance of PLIN2 lipid droplets in brain areas such as the hippocampus and cortex with the lack of lipid droplets in the healthy brain in virtually all other studies to date.

1. The authors very nicely show that the tag does not affect cell proliferation or metabolism. To assess whether the tdTom tag affects LD coat crowding, they looked at the mRNA levels of various lipases. But I would expect if there was a disruption, you might see it at the protein level (increased degradation perhaps) as opposed to changes in the mRNA. The authors should evaluate the protein levels of these lipases in control vs tdTom-Plin2 cells. By blotting for total levels, and by immunofluorescence to look at recruitment of these lipases to the LDs.
2. Figures 2K. Why didn't the authors compare the response of BD493 in the HFD to that of the control mice as they did in the previous figures. I think it would strengthen the conclusions that tdTom-PLIN2 does not change LD physiology if these control animals were included.
3. Very few studies observe LDs in brain cells, and similarly the amount of neutral lipids in the brain is very low. But the tdTom-PLIN2 signal in the brain is striking! Is it that nascent lipid droplets turnover too quickly to be detected by a neutral lipid stain? While I admit unlikely, maybe PLIN2 can detach from lipid droplets in the brain? Also, lipid droplets are rarely observed by EM either, especially in neurons. The authors should test if tdTom-PLIN2 colocalizes with BD493 in the cortex or hippocampus. With their mice, they can avoid using detergents which may be problematic in other studies. Even better if they could do CLEM to see what the lipid droplets look like that are evading other studies. And they should discuss this discrepancy in the field in more detail.
4. Figure 3G-K and Figure 5F. The authors looked at cell-type specificity of tdTom-PLIN2 expression by immunohistochemistry. In order to determine whether these lipid droplets are within the cell types, Z-stacks of these cells need to be imaged and displayed. This is a major
5. When the tdTom-PLIN2 positive cells were FACS sorted and scRNA seq was performed. What does the distribution of cell types look like in this dataset? These data could help strengthen the conclusions in comment 4 (although z-stacks are still needed!).
6. In the discussion, the authors state that FACS analysis suggests that approximately 4% of all cells in the healthy adult mouse brain have lipid droplets. But the imaging paints a different picture. The tdTom-PLIN2 is extremely abundant, and the "coloc" would claim they are in almost all cell types. Can you address this?
7. The conclusion stating that the brain has the capacity to store TAGs upon short-term HFD is based on a very subtle increase, and in some cases a non-significant trend. I recommend toning this conclusion down considerably (or do the long-term HFD).
8. For the live cell imaging experiments, why image every 5-minutes? Could the decrease in tdTom-PLIN2 staining just be photobleaching? Why not image once at different amounts of time? The speculation that lipid droplets are being consumed in ASCF due to starvation is counter to

previous studies showing that starvation increases lipid droplets (DOI: 10.1016/j.devcel.2015.01.029). This should be addressed.

Minor

1. why is there variability in Figure 1C control? Shouldn't the amount of tdTom mRNA be zero in every n measured?
2. The zoom of figure F is below, but it's spacing suggests made it seem like it was an unrelated figure. Can you move these closer together?
3. The authors use S100B to stain ependymal cells, but S100B is commonly used to stain astrocytes. Is there a better marker for ependymal cells? Otherwise, there is a risk you might pick up a specialized astrocytes such as those that communicate more with the vasculature.
4. cluster 2 microglia (plin2-positive) are discovered to show gene expression profile suggestion of activation. Is the presence of semi-activated microglia under basal conditions expected/surprising?
5. Line 336 the authors write "As LDs mainly store TAGs" in reference to the brain. What about cholesteryl esters? Is this known that LDs in the brain are TAG-rich? If yes, it would be great to add a reference here.

Reviewer #2 (Remarks to the Author):

The manuscript by Madsen et al. titled A fluorescent perilipin 2 knock-in mouse model visualizes lipid droplets in the developing and adult brain, reports the generation of lipid-droplet reporter mouse line in which perilipin 2 is tagged with tdTomato. Tagging the endogenous Plin2 gene allows for staining-free visualization of plin2-positive lipid droplets in many mouse tissues, since plin2 is ubiquitously expressed. The authors do a thorough characterization of the knock-in mouse line, including validation that the tag is properly labeling lipid droplets in expected tissues, that it is not altering the overall expression of plin2, and that the localization and number of lipid droplets in various tissues is not altered as a result of the tag. The authors also confirm that the tag does not alter any metabolic parameters and that the knock-in line responds similarly to control mice when challenged with a high-fat diet. The tdTomato-plin2 mice are then used to characterize lipid droplets in the adult and developing brain. The authors find an abundance of lipid droplets with a large range of size diversity in various regions of the adult brain. Using immunohistochemistry for different cell type markers and by crossing the tdTomato-plin2 mice with EGFP-reporter mouse lines marking neurons and astrocytes, the authors are able to further define what cell types in the brain contain plin2-positive lipid droplets. Single cell RNA sequencing was used to define different populations of microglia with varying amounts of plin2-positive lipid droplets, something that could be further extended to other cell types if desired. Lastly, analysis of embryonic brains reveal that lipid droplets are abundant in the developing mouse cortex and that in slice cultures, the lipid droplets in these embryonic mouse brain tissues can react dynamically to exogenously provided lipids.

The validation of the tdTomato-Plin2 knock-in line is very thorough and the characterization of the plin2-positive lipid droplets in the brain is compelling. The tdTomato-plin2 knock-in mouse line is a much-needed addition to the toolbox for studies of plin2 and lipid droplets in all mouse tissues and is likely to be used by researchers in all different fields of study. Additionally, the text of the manuscript was clear and easy to follow. Overall, the paper was very compelling and well executed reflecting a significant advance appropriate for publication Nature Communications

However, prior to publication, there are some issues that need to be addressed:

1. In general, throughout the manuscript, the authors give the reader the impression that all lipid droplets are labeled by perilipin 2. While plin2 is expressed ubiquitously, it is not necessarily labeling every lipid droplet in the tissues of the body, so it is important to remind readers occasionally that this is a plin2-positive lipid droplet reporter.

2. In Figure 1D, it would have been nice to see the uncropped immunoblot for PLIN2, so that it is possible to evaluate the size difference between wild-type PLIN2 and the tdTomato-PLIN2. Is the tdTom-PLIN2 blot in this figure done with the tdTomato antibody or the PLIN2 antibody?
3. For Figure 1H, please better define "metabolic potential" for the reader.
4. In Figure 1J, it was unclear why the authors only included mRNA expression for Atgl, Hsl and Mgl and not protein levels or immunofluorescence. Is it known that the mRNA expression levels vary in different metabolic states and that this influences protein levels/localization appreciably?
5. In Figure 2L, it would be nice to see liver images from the control diet as well as the high-fat diet
6. For Figure 2M, please include the blot of the untagged PLIN2, since these mice are heterozygous for the tdTomato-PLIN2 and therefore have an untagged version as well. Does the pattern look the same for the untagged PLIN2?
7. Although it was not a statistically significant difference, why was there a slight decrease in the Plin2 mRNA levels in the brain of the tdTomato-Plin2 mouse? A similar decrease was also noted for atgl, hsl and mgl expression in the brains as shown in Supplementary Figure 2. Does this suggest an issue with the sample quality of the mRNA/cDNA?
8. In Figure 3G, there is an arrow missing in the tdTom-PLIN2 image panel.
9. In Figure 4A, the experimental set-up indicates sorting for tdTomato-Plin2 positive cells before doing single cell RNA sequencing. The authors then focus on just the microglia. The text in the final paragraph of this section of the results makes it sound like the microglia were sorted as well, but I think they were identified bioinformatically. Please clarify for the reader.
10. Are the bottom image panels in Figure 5B & C, magnifications of the upper image panels? If so, please provide dotted squares to indicate the zoomed regions.
11. The data indicate that the NSPCs in vivo do not have as many lipid droplets as NSPCs in vitro. Have the lipid droplets in these cells also been examined with BODIPY? Is it possible that they have the same numbers of lipid droplets, but that in vitro, all of the lipid droplets are plin2-positive?
12. Similarly, in Figure 6G, and supplementary Figure 5, it would be nice to see wholemount images stained with BODIPY in addition to the tdTomato-Plin2 images in order to know if all of the lipid droplets are plin2-positive and or whether the high-fat diet specifically increases plin2-positive lipid droplets? Are there changes in lipid droplet size with this diet?
13. In a number of figure panels (for example, Fig. 2G, H, I, J and Fig. 6 A & C) in which the mice were fed either the standard diet or the high-fat diet, the authors chose to make two similar graphs for a measured parameter instead, one for the control mice and one for the tdTomato-Plin2 mice. It was unclear why these were not combined into one graph and it was unclear why statistics were only performed between the SD and the HFD within a genotype and no statistics were performed between genotypes.
14. On line 332, the reference to Fig 2E was a bit confusing since it was images of liver and intestine, not a description of the diets.

Reviewer #3 (Remarks to the Author):

To visualize LD dynamics in vivo, the authors generated tdTom-PLIN2 mice, in which a fluorescent protein tdTomato is knocked in at the locus encoding PLIN2, one of the LD surface proteins, and

used these mice to clarify the distribution of LD in the brain, LD dynamics in the embryonic brain, and changes in LD upon addition of high-fat diet. Although LD is recognized as a universally present organelle, the dynamics of LD in the brain found by the authors are very interestingly. The experimental methods are specific and precise, and the interpretation of the results is reasonable. However, since the presence of LD in the brain has been well known in recent years, the present data alone are not novel, and a revision in response to the comments below is necessary to fully agree with the authors' claims and support publication in Nature communications.

Major comments:

Since the PLIN2 protein may not localize to nascent (immature) LDs, the authors should use antibodies against other PLIN proteins (such as PLIN1 and PLIN3) to correlate with LDs that are labeled with tdTom-PLIN2. Similarly, the pattern of LD emergence during embryonic brain development should be compared using antibodies against other PLIN proteins and LD marker antibodies. These points should be clarified, especially since the distribution of LDs during brain development may simply be due to increased expression levels of PLIN2 protein. In addition to this, changes in the expression levels of other PLIN proteins such as PLIN1 and PLIN3 should be added to the Western blot data in Fig. 1D and Fig. 2M. Alternatively, mRNA level expression levels should be compared.

Since it is known that intracellular LD interacts with various organelles and that the interaction changes with nutritional status, the correlation between LD and other organelles (ER, mitochondria, lysosomes, etc.) in obesity and during brain development should be analyzed. In this connection, LD distribution in the brain and its correlation with other organelles should be analyzed using other methods (e.g., electron microscopy). This is because almost all data concerning LD distribution are dependent on fluorescence observations with the tdTom-PLIN2 reporter.

The amount of LD in cells is maintained in balance between synthesis and degradation, so it should be clarified whether LD distributed in the brain is degraded or not. For example, the impact of changes in physiological conditions, such as starvation-induced changes or the addition of a low-fat diet (temporarily) after the addition of a high-fat diet, on the amount of LD in vivo (liver, brain, etc.) should be elucidated.

Minor comments:

This reviewer is not willing to define a 4-month high-fat diet feeding as SHORT-TERM because even a few days of high-fat diet feeding is often defined as SHORT-TERM. It would be better to specify a specific period of time.

In Fig. 2L: it is better to include an image of the SD control.

Reviewer #4 (Remarks to the Author):

In the manuscript entitled "A fluorescent perilipin 2 knock-in mouse model visualizes lipid droplets in the developing and adult brain", Sofia Madsen et al., generated a Lipid droplet (LD) -reporter mouse by endogenously labelling the LD specific protein PLIN2 with tdTomato using CRISPR/Cas9. This tdTom-Plin2 mouse model provides a novel tool to study LDs and their dynamics. Combine this reporter mouse with scRNA-seq and live imaging, they showed that LDs are abundant in the main brain cell types of both embryonic and adulthood brain. However, the novelty of biological process appears weak and there are some concerns about bioinformatic analysis. The unique advantage of this reporter line has not been demonstrated to the most. The specific comments are listed as the following:

Major concerns:

1. Although authors mentioned that endogenous tagging of Plin2 with tdTomato does not significantly alter the mRNA expression, protein expression or the amount of LDs in NSPCs in vitro, however it appears that the protein expression of tagged and untagged Plin2 together (add up 105kDa and 51kDa forms) in tdTom-Plin2 mouse was higher than control mouse (Figure 1D). So does the protein level always correlate with RNA expression? In figure 2M, the author only examined the bigger form or PLIN2.
2. The Plin2-tdTomato is heterogeneously expressed in the mice. The reviewer wondered if the expression of TdTomato always dictate the expression of Plin2 (it sees so in Figure 2D). In figure 4, besides the cell clustering based on the expression of TdTomato, the expression of Plin2 could also be displayed. In this case, cluster 2 has higher tDTomato positive cells and shall expression more Plin2 and the DEG analysis (Figure 4G) may reflect this trend.
3. It is interesting that 95 % of the NSPCs expressing PLIN2 also expressed tdTom-PLIN2 and 80% of all LDs expressing PLIN2 showed colocalization with the endogenous tdTom-PLIN2. If the author used the classical immunofluorescent staining (lipid dyes OilRedO or BD493) to identified the LDs, what would be the colocalization in the proportion of tdTom-PLIN2?
4. The authors found some DEGs in different clusters of microglia (Figure 4C,F,G), what are their functional relevance? Are there some key factors to regulate the transformation from different microglia states?
5. In Figure 6 the data showed minor differences and the conclusion "Short-term high fat diet increases the total TAG levels in the brain" is not convincing. Besides, classic immunofluorescent staining lipid dyes BD493 in SD and HFD should be added to compare the LD changes in whole mounts lateral ventricle wall.

Minor Concerns:

1. The schematic illustration of genetic process in Figure 1A is not clear to understand the generation of tdTom-Plin2 mouse, even was referred to the methods.
2. Authors should add the quality assessment of scRNA-seq, such as detected gene number, mapping ratio and mitochondrial genes percentage.

Point by point reply NCOMMS-22-29252-T

We would like to thank the reviewers for their overall appreciation of our work and for their suggestions to improve the manuscript. We have now extensively revised our initial manuscript and have added the following key results:

- 1.) A **complete histopathological analysis** (conducted blindly by a certified pathologist) of 3 tdTom-Plin2 het vs 3 tdTom-Plin2 wt littermates, confirms that there is **no alteration due to the endogenous tagging** in any of the tissues analyzed. These results are in the new supplementary Figure 1.
- 2.) We have **established a protocol** which allows us to **visualize LDs by classical staining methods using antibodies against PLIN2 and different lipid dyes**. We have systematically compared tissue pre-treatment and staining conditions (cryosections versus vibratome, as well as different detergent concentrations). Through this, we can show that **LDs are indeed as abundant in Ctrl mice as we report with the tdTom-Plin2 reporter mouse**. We furthermore demonstrate that the **classical lipid dye BODIPY493 shows limited efficacy in brain tissue**, whereas alternative **lipid dyes** commercially available reveal LDs to the same extent as observed in the tdTom-Plin2 reporter mouse. Furthermore, these dyes **co-localize with the endogenous tdTomato signal, confirming the authenticity of the identified LDs**. Given the extensive contributions from other lab members not involved in this manuscript, and as we want to make all the details of this work available, we have discussed with the **handling editor at Nature Communications** how to best proceed. **We agreed to consolidate these findings into a separate, more technical manuscript and refer to it in the revised version of this manuscript**. We have included it as supplementary material with the current revision and **will make it available on BioRxiv** upon publication of this manuscript if the reviewers agree. For your convenience, we have added the key findings in the detailed responses below.
- 3.) We have **the 4-week HFD treatment in a new cohort of mice** to time-match the sacrifice of Ctrl and tdTom-Plin2 mice. Initially, cohorts were time-matched based on genotype, hence comparisons of the diet effect were conducted within each genotype. The addition of this new cohort enhances the robustness of the data. We show that **both the Ctrl and tdTom-Plin2 reporter mice exhibit similar responses, with increased LDs in the liver after HFD**. Additionally, we have adapted the liver analysis to incorporate whole liver sections. These updated results are now integrated into the revised Figure 2.
- 4.) We have performed a proteomics analysis comparing Ctrl and tdTom-Plin2 and NSPCs and **show that all the major proteins involved in LD biogenesis and turnover are not altered due to the tagging approach**. These data are now included in the revised Figure 1/S1.

We have also addressed all other points raised by the Reviewers, as detailed below.

Reviewer #1 (Remarks to the Author):

Lipid droplets are important lipid storage organelles that play a critical role in cell physiology and consequently many diseases. Here, Madsen et al. developed a transgenic mouse line to label lipid droplets. They endogenously tagged the lipid droplet resident protein PLIN2 with tdTomato. They show that their tagging strategy does not disrupt PLIN2 function as they observe no major effects on lipid metabolism. They demonstrate the usefulness of their model system in different organ systems (liver and the brain) and in fixed tissue, cultured cells, and ex vivo brain slices. Overall, I think there is great value in developing this mouse line. It will be useful for biologists across different physiological systems. Particularly in the brain, where the role of lipid droplets is a rapidly growing field. I think the study is well written and for the most part, well-controlled.

We thank this reviewer for her/his overall appreciation and for sharing the view that this mouse model has great value.

However, there are a few technical issues that need to be addressed first. Notably, Z-stacks need to be performed to conclude the cell-types that contain the lipid droplets. And the authors need to better reconcile the sheer abundance of PLIN2 lipid droplets in brain areas such as the hippocampus and cortex with the lack of lipid droplets in the healthy brain in virtually all other studies to date.

We agree with the reviewer regarding the abundance of LDs observed with this model is striking and much more than in all other studies to date in the healthy brain. To verify that these findings reflect the status of LD abundance in wildtype mice, we have developed a protocol to figure out why conventional LD staining methods work so poorly on mouse brain tissue. Through systematic testing involving various fixation techniques, tissue treatments (freezing or not), permeabilization steps, and utilization of several commercially available lipid droplet dyes, we are pleased to show that we can now detect a similar number of LDs in Ctrl mice brain tissue, confirming the findings in the tdTom-Plin2 mouse. We were also surprised to find that Bodipy493/503 (BD493), a commonly used LD stain, does not work well in brain tissue, whereas other LD dyes such as LipiDye II and LipidSpot610 show a similar pattern as Plin2 staining. With these convincing stainings in wildtype brain tissue, we conclude that the reason for the lack of LDs in previous studies must be due to technical limitations. Due to the extensive nature of establishing LD staining in mouse brain tissue, we have decided to present the details a separate technical paper. This manuscript provides all the details for successful staining of LDs in healthy mouse brain tissue, using a Plin2 antibody and different lipid dyes. We have already shared this protocol with other collaborators, who were successful in revealing LDs with our method. To link it to what we see with the tdTom-Plin2 mouse model, we included brain tissue section from this model. **The manuscript has been attached separately to this revision and we will make it available on BioRxiv for proper reference, in case this manuscript is accepted for publication.** We have added here the key findings so that the Reviewer's do not necessarily have to go through the additional manuscript. **For copy-right issues, the Figures have been removed from this point by point reply. They can be found in the manuscript Petrelli, Rey et al, BioRxiv 2024.**

This figure (**Figure 2 in the separate manuscript**) shows that BD493 works poorly in brain tissue of adult WT mice, whereas two other commercially available dyes (LipiDye II and LipidSpot) reveal a large number of LDs in healthy WT brains.

Figure removed here for copy-right issues, it can be found in the manuscript Petrelli, Rey et al, BioRxiv 2024

Redacted

This figure (**Figure 3 in the separate manuscript**) shows that a specific protocol of immunohistochemistry against PLIN2 reveals a large number of LDs in brain tissue of adult WT mice, but the detection depends on the tissue pretreatment.

Figure removed here for copy-right issues, it can be found in the manuscript Petrelli, Rey et al, BioRxiv 2024

Redacted

This figure (**Figure 5 in the separate manuscript**) shows that LipidSpot488 and LipiDylI signal colocalize with endogenous tdTomato signal in tdTom-Plin2 reporter mice, confirming that the red structures are indeed LDs. It also shows that this colocalization is not possible in the WT brain, as the staining procedure seems to disrupt the dye staining.

Figure removed here for copy-right issues, it can be found in the manuscript Petrelli, Rey et al, BioRxiv 2024

Redacted

Regarding the comment about Z-Stacks: All the images shown in this manuscript are indeed confocal images with Z-Stacks. We show maximum intensity projections over the volumes of the cells shown. We have now clarified this in each figure.

1. The authors very nicely show that the tag does not affect cell proliferation or metabolism. To assess whether the tdTom tag affects LD coat crowding, they looked at the mRNA levels of various lipases. But I would expect if there was a disruption, you might see it at the protein level (increased degradation perhaps) as opposed to changes in the mRNA. The authors should evaluate the protein levels of these lipases in control vs tdTom-Plin2 cells. By blotting for total levels, and by immunofluorescence to look at recruitment of these lipases to the LDs.

We appreciate the reviewer's emphasis on the importance of protein levels in our study. To provide a comprehensive and quantitative analysis, we have decided to do proteomics comparing Ctrl and TdTom-Plin2 het NSPCs. This approach allows a quantitative measure of all LD-related proteins, which does not rely on semi-quantitative techniques such as western blotting. We opted for proteomics due to challenges with unspecific bands observed in western blotting. Our analysis revealed that 99.52% of all detected proteins remained unchanged. Moreover, proteins involved in TAG and LD biosynthesis as well as LD breakdown and fatty acid beta-oxidation are unchanged. **These data are now included in the revised Figure 1 and revised supplementary Figure 1.**

2. Figures 2K. Why didn't the authors compare the response of BD493 in the HFD to that of the control mice as they did in the previous figures. I think it would strengthen the conclusions that tdTom-PLIN2 does not change LD physiology if these control animals were included.

We agree with the Reviewer. The reason why we have kept all the analyses between the genotypes separate and only compared the effect of the diets within one genotype was the design of the study. We had two cohorts of animals (Ctrl and TdTom-Plin2 mice) with 2 different diets. We had grouped and perfused them according to the genotype, distributed over an entire day. (One done in the morning, one in the afternoon). To make sure that the time of perfusion does not affect the analysis, especially the LDs in the liver, we have now repeated a cohort of mice, this time pairing in each cage a Ctrl and tdTom-Plin2 littermate with the same diet, and perfusing them in a staggered experiment over days, always at the same time of the day. As we have obtained the same results with this new cohort, we are now confident to pool the data of both cohorts and to directly compare Ctrl and tdTom-Plin2 mice and the effect of diets, using 2 Way ANOVAs and posthoc testing.

Additionally, we consulted experts in liver physiology, and modified our approach to consider entire liver section due the heterogeneous distribution of LDs, especially with a short term HFD. Following re-staining and re-analysis of all the livers, we observed consistent responses in both Ctrl and tdTom-Plin2 mice, including increased body fat and liver LDs after 4 weeks of HFD. These findings confirm that the tag does not influence LDs dynamics when challenged with a 4 weeks HFD. These new results are now **included in the revised Figure 2.**

3. Very few studies observe LDs in brain cells, and similarly the amount of neutral lipids in the brain is very low. But the tdTom-PLIN2 signal in the brain is striking! Is it that nascent lipid droplets turnover too quickly to be detected by a neutral lipid stain? While I admit unlikely, maybe PLIN2 can detach from lipid droplets in the brain? Also, lipid droplets are rarely observed by EM either, especially in

neurons. The authors should test if tdTom-PLIN2 colocalizes with BD493 in the cortex or hippocampus. With their mice, they can avoid using detergents which may be problematic in other studies. Even better if they could do CLEM to see what the lipid droplets look like that are evading other studies. And they should discuss this discrepancy in the field in more detail.

We absolutely agree with the Reviewer and have addressed this issue now extensively! Please see our reply to the general comment above. With our modified protocol, we are able to reveal LDs in the healthy wildtype mouse brain to a similar extent as in the tdTom-Plin2 mouse model. Showing that the lack of detection in other studies might be of technical nature. We also show that the endogenous tdTom-Plin2 positive structures co-localize with two commercially available lipid dyes. We now refer in this revision to our separate manuscript and have added a paragraph in the discussion to discuss this discrepancy in the field in more details.

4. Figure 3G-K and Figure 5F. The authors looked at cell-type specificity of tdTom-PLIN2 expression by immunohistochemistry. In order to determine whether these lipid droplets are within the cell types, Z-stacks of these cells need to be imaged and displayed.

This might be a misunderstanding. All images presented are acquired using high-resolution confocal microscopy. In order to visualize the entire cell, we have z-projected the stacks over the volume of the specific cell types. We have now clarified this in both the figure legend and in the method.

5. When the tdTom-PLIN2 positive cells were FACS sorted and scRNA seq was performed. What does the distribution of cell types look like in this dataset? These data could help strengthen the conclusions in comment 4 (although z-stacks are still needed!).

We agree with the Reviewer that the FACS and scRNA experiment could have been informative about the distribution of cell types containing LDs. Unfortunately, our current protocol resulted in the majority of good sequencing reads coming from microglia and did not give a good representation of other cell types. As stated in the initial manuscript, this might be due to the lengthy protocol, with creating single cell suspension, FACS followed by single bead production using the 10x Genomix platform. While it would be great to use flow cytometry to sample all the cell types in the brain, it is evident that we need to refine our protocol to ensure equal representation of all cell types. We have now clarified in the manuscript that majority of good sequencing reads came from microglia precluded us from making generalized statements about the presence of LDs in different cell types.

6. In the discussion, the authors state that FACS analysis suggests that approximately 4% of all cells in the healthy adult mouse brain have lipid droplets. But the imaging paints a different picture. The tdTom-PLIN2 is extremely abundant, and the “coloc” would claim they are in almost all cell types. Can you address this?

We agree with the Reviewer that the calculated number of cells containing LDs from the FACS experiments were surprisingly low and do not match with what we observe with imaging. As we

cannot control whether the total cells in the FACS experiments were actually a true representation of all brain cells due to the lengthy procedure to prepare single cell suspensions for FACS, we have now removed this graph. In the experiments where we have a GFP reporter (Aldh111-GFP, Thy1-GFP and NesGFP), we can express the % of cells containing LDs in a reliable way, as we see the GFP population and just distinguish which of those are also tdTomato positive. These numbers are indeed much higher (up to 35% for astrocytes for instance). However, as we do not know how many droplets (and which size of the droplets) are required for detection by FACS, we cannot directly compare the sections with the results from the FACS.

7. The conclusion stating that the brain has the capacity to store TAGs upon short-term HFD is based on a very subtle increase, and in some cases a non-significant trend. I recommend toning this conclusion down considerably (or do the long-term HFD).

We agree with the Reviewer regarding the small effects observed in response to the 4 week HFD. Despite this, we still find it remarkable that such a relatively short duration of HFD resulted in detectable differences in TAG levels, as measured by lipidomics and in LD coverage, as indicated by the tdTom-Plin2 reporter mouse. (When combining the results of Ctrl and tdTom-Plin2 mice and running a two-way ANOVA, with posthoc tests, both HFD groups are significantly increased). However, in the revised version, we have toned this statement down, pointing out that the differences are small, and have added a statement in the discussion that a long-term HFD would allow to see if LDs and TAGs would show a bigger increase.

8. For the live cell imaging experiments, why image every 5-minutes? Could the decrease in tdTom-PLIN2 staining just be photobleaching? Why not image once at different amounts of time? The speculation that lipid droplets are being consumed in ASCF due to starvation is counter to previous studies showing that starvation increases lipid droplets (DOI: 10.1016/j.devcel.2015.01.029). This should be addressed.

We had initially imaged every 5 minutes to get an idea about the motility and fluctuations of LDs and we did not alter the imaging procedure. The reviewer points out correctly that photobleaching could be an issue. Thus, we have now repeated the live-imaging and imaged only every 30min. We still see the same decrease in LDs over time as with the 5min imaging paradigm. This suggests that this decrease is most likely not due to photobleaching. The speculation that LDs are consumed came from the observation that with prior oleic acid loading, the decrease is less pronounced. We have added the new data in the same supplementary figure (Fig. S7) and have added in the discussion part that we cannot exactly tell why the number of LDs reduce over time. We are currently following this up in a new project addressing the role of LDs during development, so we hope to provide more insights in the future about their actual role.

Minor

1. why is there variability in Figure 1C control? Shouldn't the amount of tdTom mRNA be zero in every n measured?

Yes, ideally it should be zero, but as we are looking at Ct values, the tdTom probe is still giving some very late reads (Ct 39, as does the water only control). When calculating the dCT from these values, we do not technically end up with 0.

2. The zoom of figure F is below, but it's spacing suggests made it seem like it was an unrelated figure. Can you move these closer together?

We have done this as suggested.

3. The authors use S100B to stain ependymal cells, but S100B is commonly used to stain astrocytes. Is there a better marker for ependymal cells? Otherwise, there is a risk you might pick up a specialized astrocytes such as those that communicate more with the vasculature.

S100B is a commonly used marker for staining ependymal cells (Chiasson et al., 1999). The ependymal cells line the ventricles and typically form the first layer, while mature astrocytes are less frequently observed in this location. However, GFAP+ S100B- astrocytes and GFAP+ NSCs can be found nearby the ventricle. Additionally, the distinct morphology of ependymal cells and mature astrocytes enables easy identification using S100B staining. Furthermore, we have verified the presence of LDs in ependymal cells using wholemount preparations stained with beta catenin, where LD-containing ependymal cells are clearly visible (Figure 3D). Moreover, we have identified the presence of LDs in astrocytes using glutamine synthetase (GS), a characteristic marker for astrocytes (Figure 3L).

4. cluster 2 microglia (plin2-positive) are discovered to show gene expression profile suggestion of activation. Is the presence of semi-activated microglia under basal conditions expected/surprising?

We want to clarify that while there are some shared features between the gene expression signature of Plin2-positive microglia and the DAM signature, they are not identical. Interestingly, the Plin2-positive microglia also exhibit elevated expression levels of many "housekeeping" genes that are typically downregulated in DAM, indicating distinct transcriptional profile. DAM signature microglia have not been found in the healthy brain. We think that what is indeed surprising is that the presence of LDs goes along with a specific gene expression signature. However, we cannot conclude from the gene expression signature to the function of these Plin2-positive microglia. Further studies will show whether this LD-positive population in the healthy brain changes with aging or disease.

5. Line 336 the authors write “As LDs mainly store TAGs” in reference to the brain. What about cholesteryl esters? Is this known that LDs in the brain are TAG-rich? If yes, it would be great to add a reference here.

We have changed this line to “As LDs mainly store TAGs and cholesterol esters”, as we do not know if the LDs we see contain more TAGs or more cholesterol esters.

Reviewer #2 (Remarks to the Author):

The manuscript by Madsen et al. titled A fluorescent perilipin 2 knock-in mouse model visualizes lipid droplets in the developing and adult brain, reports the generation of lipid-droplet reporter mouse line in which perilipin 2 is tagged with tdTomato. Tagging the endogenous Plin2 gene allows for staining-free visualization of plin2-positive lipid droplets in many mouse tissues, since plin2 is ubiquitously expressed. The authors do a thorough characterization of the knock-in mouse line, including validation that the tag is properly labeling lipid droplets in expected tissues, that it is not altering the overall expression of plin2, and that the localization and number of lipid droplets in various tissues is not altered as a result of the tag. The authors also confirm that the tag does not alter any metabolic parameters and that the knock-in line responds similarly to control mice when challenged with a high-fat diet. The tdTomato-*plin2* mice are then used to characterize lipid droplets in the adult and developing brain. The authors find an abundance of lipid droplets with a large range of size diversity in various regions of the adult brain. Using immunohistochemistry for different cell type markers and by crossing the tdTomato-*plin2* mice with EGFP-reporter mouse lines marking neurons and astrocytes, the authors are able to further define what cell types in the brain contain plin2-positive lipid droplets. Single cell RNA sequencing was used to define different populations of microglia with varying amounts of plin2-positive lipid droplets, something that could be further extended to other cell types if desired. Lastly, analysis of embryonic brains reveal that lipid droplets are abundant in the developing mouse cortex and that in slice cultures, the lipid droplets in these embryonic mouse brain tissues can react dynamically to exogenously provided lipids.

The validation of the tdTomato-*Plin2* knock-in line is very thorough and the characterization of the plin2-positive lipid droplets in the brain is compelling. The tdTomato-*plin2* knock-in mouse line is a much-needed addition to the toolbox for studies of plin2 and lipid droplets in all mouse tissues and is likely to be used by researchers in all different fields of study. Additionally, the text of the manuscript was clear and easy to follow. Overall, the paper was very compelling and well executed reflecting a significant advance appropriate for publication Nature Communications

We thank the Reviewer for her/his overall appreciation of the manuscript and for sharing the view that this LD mouse model is useful for a large field of researcher.

However, prior to publication, there are some issues that need to be addressed:

1. In general, throughout the manuscript, the authors give the reader the impression that all lipid droplets are labeled by perilipin 2. While plin2 is expressed ubiquitously, it is not necessarily labeling every lipid droplet in the tissues of the body, so it is important to remind readers occasionally that this is a plin2-positive lipid droplet reporter.

We agree that it is important to make clear that this is a *Plin2* reporter mouse. In response, we have now revised the scheme in the updated Figure 1 to emphasize this aspect more clearly, and have mentioned it several times as a reminder throughout the results section. We have also added a separate paragraph pointing this out in the discussion

2. In Figure 1D, it would have been nice to see the uncropped immunoblot for PLIN2, so that it is possible to evaluate the size difference between wild-type PLIN2 and the tdTomato-PLIN2. Is the tdTom-PLIN2 blot in this figure done with the tdTomato antibody or the PLIN2 antibody?

We have added the uncropped immunoblot to a separate supplementary figure containing all the original western blots shown in this manuscript (Suppl. Figure 8). To reveal the unlabeled and labelled PLIN2, we have used a PLIN2 antibody.

3. For Figure 1H, please better define “metabolic potential” for the reader.

We have provided a more detailed definition of this term.

4. In Figure 1J, it was unclear why the authors only included mRNA expression for Atgl, Hsl and Mgl and not protein levels or immunofluorescence. Is it known that the mRNA expression levels vary in different metabolic states and that this influences protein levels/localization appreciably?

We agree with the Reviewer regarding the importance of protein levels over mRNA levels. To provide a comprehensive and quantitative analysis, we performed proteomics comparing Ctrl and TdTom-Plin2 het NSPCs. This allows a quantitative measure of all LD-related proteins, which does not rely on semi-quantitative techniques such as western blotting. We opted for proteomics due to challenges with unspecific bands observed in western blotting. Our analysis revealed that 99.52% of all detected proteins remained unchanged. Moreover, proteins involved in TAG and LD biosynthesis as well as LD breakdown and fatty acid beta-oxidation are unchanged. **These data are now included in the revised Figure 1 and revised supplementary Figure 1.**

5. In Figure 2L, it would be nice to see liver images from the control diet as well as the high-fat diet

We did show the control diet images in Figure 2E, but we see that this might not have been clear enough and a side-by-side comparison is easier for the reader. Consequently, **We have now substantially revised Figure 2. The updated figure now incorporates new data from Ctrl mice and both Ctrl and HFD liver examples from both Ctrl and tdTom-Plin2 mice**, ensuring a comprehensive comparison for improved clarity.

6. For Figure 2M, please include the blot of the untagged PLIN2, since these mice are heterozygous for the tdTomato-PLIN2 and therefore have an untagged version as well. Does the pattern look the same for the untagged PLIN2?

The western blot shown in Figure 2M was done with the tdTomato antibody, resulting in the detection of only saw the tagged version. We have now done additional WBs using the PLIN2 antibody revealing that both the tagged and the untagged versions follow the same pattern, namely a significant increase in PLIN2 after HFD. We have now extensively revised Figure 2 and relocated the Western blots to the revised Supplementary Figure 2, showing both the tdTomato antibody and

the Plin2 antibody. Additionally, all western blots have been included in an uncropped version in Suppl. Figure 8.

7. Although it was not a statistically significant difference, why was there a slight decrease in the Plin2 mRNA levels in the brain of the tdTomato-Plin2 mouse? A similar decrease was also noted for atgl, hsl and mgl expression in the brains as shown in Supplementary Figure 2. Does this suggest an issue with the sample quality of the mRNA/cDNA?

The samples were processed in parallel, so we think we can rule out sample quality problems. We do not know why there is a slight decrease.

8. In Figure 3G, there is an arrow missing in the tdTom-PLIN2 image panel.

Thank you for pointing this out, we have added the arrow to the panel.

9. In Figure 4A, the experimental set-up indicates sorting for tdTomato-Plin2 positive cells before doing single cell RNA sequencing. The authors then focus on just the microglia. The text in the final paragraph of this section of the results makes it sound like the microglia were sorted as well, but I think they were identified bioinformatically. Please clarify for the reader.

The Reviewer is accurate in pointing out that the sorting was done for tdTomato-positive Plin2 cells and microglia were subsequently identified bioinformatically. As also explained in the reply to Reviewer 1, point 5, microglia were the cell types with the best quality reads, whereas other cell types were poorly reflected in the single cell RNAseq. As the protocol to isolate cells, sort them and then do single beads for 10x Genomics is lengthy, we would need to optimize the survival of all brain cells to get a more complete picture. We have now provided this further clarification on this aspect in the manuscript within the result and discussion sections.

10. Are the bottom image panels in Figure 5B & C, magnifications of the upper image panels? If so, please provide dotted squares to indicate the zoomed regions.

No, these are not magnifications but separate images.

11. The data indicate that the NSPCs in vivo do not have as many lipid droplets as NSPCs in vitro. Have the lipid droplets in these cells also been examined with BODIPY? Is it possible that they have the same numbers of lipid droplets, but that in vitro, all of the lipid droplets are plin2-positive?

We agree with the reviewer, that the abundance of LDs in NSPCs in vivo appears lower compared to their in vitro counterparts. However, our new staining method (explained in details in the answers to reviewer 1) shows perfect co-localization between LDs revealed by a lipid dye in vivo and the endogenous tdTom-Plin2 signal. This finding makes this hypothesis of reduced LD presence in vivo less plausible. Instead, we speculate that the variance may be attributed to differences in metabolic

environment (high nutrients in vitro) and cellular activity states. We want to follow this up in more details in the future.

12. Similarly, in Figure 6G, and supplementary Figure 5, it would be nice to see wholemount images stained with BODIPY in addition to the tdTomato-Plin2 images in order to know if all of the lipid droplets are plin2-positive and or whether the high-fat diet specifically increases plin2-positive lipid droplets? Are there changes in lipid droplet size with this diet?

We have successfully developed a protocol to simultaneously visualize the lipid core (with lipid dyes) and the tdTomato signal. (Please refer to the detailed reply and figures in the response to Reviewer 1). These stainings show excellent co-localization of tdTom-Plin2 with LipidSpot488, even in the SVZ (see Figure Page 5). Consequently, we have high confidence that our endogenous reporter line effectively detects the majority of LDs.

13. In a number of figure panels (for example, Fig. 2G, H, I, J and Fig. 6 A & C) in which the mice were fed either the standard diet or the high-fat diet, the authors chose to make two similar graphs for a measured parameter instead, one for the control mice and one for the tdTomato-Plin2 mice. It was unclear why these were not combined into one graph and it was unclear why statistics were only performed between the SD and the HFD within a genotype and no statistics were performed between genotypes.

The reason behind conducting separate analyses for each genotype and only comparing the effects of the diets within each genotype was the design of the study. We initially had two cohorts of animals (Ctrl and TdTom-Plin2 mice) with 2 different diets. These cohorts were grouped and perfused based on genotype, distributed over an entire day. (One done in the morning, one in the afternoon). To ensure that the timing of perfusion did not affect the analysis, especially the LDs in the liver, we repeated the experiment using a new cohort of mice. This time, we paired a Ctrl and tdTom-Plin2 littermate from each cage, both fed the same diet, and perfusing them in a staggered experiment over days, always at the same time of the day. The consistent results obtained from this new cohort, have provided confidence in pooling the data of both cohorts and to directly compare Ctrl and tdTom-Plin2 mice and the effect of diets, using 2 Way ANOVAs and posthoc testing. These data show that both Ctrl and tdTom-Plin2 mice react very similarly, with a significant increase in body fat and increased liver LDs after 4 weeks HFD, suggesting that the tag does also not influence LDs when challenged with a 4 weeks HFD. These new results are now **included in the revised Figure 2**.

14. On line 332, the reference to Fig 2E was a bit confusing since it was images of liver and intestine, not a description of the diets.

We have modified this accordingly to make it clearer.

Reviewer #3 (Remarks to the Author):

To visualize LD dynamics in vivo, the authors generated tdTom-PLIN2 mice, in which a fluorescent protein tdTomato is knocked in at the locus encoding PLIN2, one of the LD surface proteins, and used these mice to clarify the distribution of LD in the brain, LD dynamics in the embryonic brain, and changes in LD upon addition of high-fat diet. Although LD is recognized as a universally present organelle, the dynamics of LD in the brain found by the authors are very interestingly. The experimental methods are specific and precise, and the interpretation of the results is reasonable. However, since the presence of LD in the brain has been well known in recent years, the present data alone are not novel, and a revision in response to the comments below is necessary to fully agree with the authors' claims and support publication in Nature communications.

We thank the Reviewer for her/his general appreciation of our study. However, we would like to point out that while the presence of LDs in the brain has garnered considerable attention recently, it is noteworthy that, to our knowledge, no prior study has shown the abundance of LDs in the healthy developing and adult brain. We attribute this to the limitations of conventional staining approaches, which we have successfully overcome with the development of this endogenous LD-reporter mouse. This novel approach has enabled us to shed a new light on LDs in the brain, marking a significant advancement in understanding of brain lipid metabolism.

Major comments:

Since the PLIN2 protein may not localize to nascent (immature) LDs, the authors should use antibodies against other PLIN proteins (such as PLIN1 and PLIN3) to correlate with LDs that are labeled with tdTom-PLIN2. Similarly, the pattern of LD emergence during embryonic brain development should be compared using antibodies against other PLIN proteins and LD marker antibodies. These points should be clarified, especially since the distribution of LDs during brain development may simply be due to increased expression levels of PLIN2 protein. In addition to this, changes in the expression levels of other PLIN proteins such as PLIN1 and PLIN3 should be added to the Western blot data in Fig. 1D and Fig. 2M. Alternatively, mRNA level expression levels should be compared.

We agree with the Reviewer that the labelling of the LDs in our system depends on the presence of PLIN2. As explained in detail in the general section of this point-by-point reply and **as described in details in the reply to reviewer 1**, we have invested significant time to establishing a “classical” staining protocol using antibodies and lipid dyes to show that tdTom-Plin2 LDs reflect the majority of the LDs. Through quantitative analysis, we have determined the co-labeling of lipid-dye positive LDs with the tdTomato signal, revealing that with the most effective lipid dyes (LipidSpot and LipiDye II), ~90% of the tdtomato positive LDs are also positive for the lipid dyes (see below). We also show an excellent co-localization of LipidSpot and LipiDye II with the tdTom-Plin2 signal in vivo (see Page 5), so we do not think that we are missing LDs that would be labeled by other perilipin family members and not contain Plin2 in NSPCs and brain tissue.

This figure (Fig. 1 in the separate manuscript) shows that LipidSpot and LipiDye II signal co-localizes with endogenous tdTomato signal in tdTom-Plin2 reporter mice, confirming that the red structures are indeed LDs.

Figure removed here for copy-right issues, it can be found in the manuscript Petrelli, Rey et al, BioRxiv 2024

These data also reflect the expression pattern of Perilipin family members in NSPCs (Ramosaj, Madsen et al. Nature Communications 2021, graph copied from the publication for the convenience of the reviewer), showing that Plin2 is by far the most expressed Perilipin family member protein in NSPCs.

As suggested, we have added a PLIN3 western blot, showing no change between the tdTom-Plin2 and Ctrl NSPCs (revised Figure S1A). Furthermore, we have performed proteomics analyses, showing no changes in the protein levels of proteins involved in the TAG and LD buildup, and in LD breakdown and fatty acid beta-oxidation (revised Fig. 1 J). Of all the Perilipin family members, only Plin2 is sufficiently expressed to be detectable by the proteomic approach (Revised Suppl. Fig. 1D).

We are not sure what the reviewer wanted to point out with the statement “*since the distribution of LDs during brain development may simply be due to increased expression levels of PLIN2 protein*”. As PLIN2 is a specific LD coat protein, we do think that its expression is reflecting LDs. With the staining protocols established, we show that tdTom-Plin2 signal indeed shows LDs (see Page 5). We have now also verified this in the developing brain, shown below as an example.

This figure shows a cortical region of a tdTom-Plin2 embryonic brain (E14.5) stained with LipidSpot488. It shows that the majority of the tdTomato-positive structures are also positive for LipidSpot488. (A: overview, B: higher magnification of the box).

Since it is known that intracellular LD interacts with various organelles and that the interaction changes with nutritional status, the correlation between LD and other organelles (ER, mitochondria, lysosomes, etc.) in obesity and during brain development should be analyzed. In this connection, LD distribution in the brain and its correlation with other organelles should be analyzed using other methods (e.g., electron microscopy). This is because almost all data concerning LD distribution are dependent on fluorescence observations with the tdTom-PLIN2 reporter.

We agree with the reviewer suggestion regarding the correlation of LDs with other organelles, which indeed could provide valuable insights. However, we believe that such analysis extends beyond the scope of the current manuscript. The proposed analyses of studying correlations of LDs with ER, mitochondria, lysosomes, etc. both in the context of obesity and brain development, particularly using electron microscopy would take several years to be completed. While we recognize the importance of understanding inter-organelle interactions, we intend to pursue this avenue in future endeavors involving this mouse model.

The amount of LD in cells is maintained in balance between synthesis and degradation, so it should be clarified whether LD distributed in the brain is degraded or not. For example, the impact of changes in physiological conditions, such as starvation-induced changes or the addition of a low-fat

diet (temporarily) after the addition of a high-fat diet, on the amount of LD in vivo (liver, brain, etc.) should be elucidated.

The dynamics of LDs in the brain upon starvation-induced changes or the addition of a low-fat diet would be extremely interesting to study. The advantage of our novel mouse model is that offers the potential for such investigations, including the possibility of live-imaging (2-photon with a cranial window). We are actively exploring collaborations in this direction, as we are interested in better understanding whether or not LDs change in physiological conditions. However, such studies will require several years. The primary focus of this manuscript was to introduce a novel tool for studying LDs in a dynamic manner. We have provided proof-of-concept applications, such as observing the effects diet change, cell sorting based on LD content, live imaging. We plan to address numerous remaining questions regarding the role of LDs in the brain in the coming years.

Minor comments:

This reviewer is not willing to define a 4-month high-fat diet feeding as SHORT-TERM because even a few days of high-fat diet feeding is often defined as SHORT-TERM. It would be better to specify a specific period of time.

We have specified throughout the manuscript text, in the figure scheme and in the method part that we used a 4 WEEK high-fat diet. As this is relatively short, we had called this a short-term high fat diet. The Reviewer seemed to have misread the weeks for months. We have now tried to make this clear by adding many times “4 week HFD”.

In Fig. 2L: it is better to include an image of the SD control.

We did show the control diet images in Figure 2E, but agree that a side-by-side comparison is easier for the reader. We have modified Figure 2 accordingly (including also new data from the Ctrl mice) and show now both SD and HFD liver examples of Ctrl and tdTom-Plin2 mice.

Reviewer #4 (Remarks to the Author):

In the manuscript entitled “A fluorescent perilipin 2 knock-in mouse model visualizes lipid droplets in the developing and adult brain”, Sofia Madsen et al., generated a Lipid droplet (LD) -reporter mouse by endogenously labelling the LD specific protein PLIN2 with tdTomato using CRISPR/Cas9. This tdTom-Plin2 mouse model provides a novel tool to study LDs and their dynamics. Combine this reporter mouse with scRNA-seq and live imaging, they showed that LDs are abundant in the main brain cell types of both embryonic and adulthood brain. However, the novelty of biological process appears weak and there are some concerns about bioinformatic analysis. The unique advantage of this reporter line has not been demonstrated to the most. The specific comments are listed as the following:

We regret that the Reviewer does not appreciate the unique advantage of this model. As stated in the manuscript, to our knowledge, this is the first endogenous mammalian LD reporter line, which opens up new avenues for studying LDs and their dynamics in ways that were not previously possible. We agree that while we have pointed out useful applications (such as sorting cells based on LD content, in combination with other reporter lines or followed by scRNA seq, live imaging of embryonic brain slices etc.), we have not gone into each detail for the different applications. As a thorough characterization of a novel mouse model is in our eyes key for its validity, we wanted to point out the overall possibility this novel line offers to the large LD-community. Since we added this manuscript to BioRxiv, we have received a lot of interest from the LD-community and have started several collaborations. This shows us that at least for the LD community, this model is of great interest.

Major concerns:

1. Although authors mentioned that endogenous tagging of Plin2 with tdTomato does not significantly alter the mRNA expression, protein expression or the amount of LDs in NSPCs in vitro, however it appears that the protein expression of tagged and untagged Plin2 together (add up 105kDa and 51kDa forms) in tdTom-Plin2 mouse was higher than control mouse (Figure 1D). So does the protein level always correlate with RNA expression? In figure 2M, the author only examined the bigger form or PLIN2.

We understand the concern of the Reviewer. However, we are not sure if simply “adding” intensities of different sized protein bands on a western blot to make a statement about the total amount of protein. We show that the endogenous PLIN2 band is less expressed in NSPCs from tdTom-Plin2 mice compared to the control, and that the tagged PLIN2 is only detectable in tdTom-Plin2 mice. To provide a more comprehensive analysis of protein levels, we have included proteomics data of tdTom-Plin2 and Ctrl NSPCs. These data reveal that PLIN2 protein levels are identical (as this is based on peptide detection, the PLIN2 detection comes from both the tagged and untagged version),

indicating that the tag does not alter PLIN2 protein levels. **These data have been added to the revised Figure 1 and supplementary Figure 1.**

In figure 2M, we wanted to show the usefulness of the tdTomato to detect changes in LDs upon a HFD, thus we looked at the tdTomato signal. We have now added Western blots with the PLIN2 antibody. We show that both the tagged and untagged version increase with the HFD. As we have substantially revised this figure, the western blots are now in the revised Supplementary Figure 2.

2. The Plin2-tdTomato is heterogeneously expressed in the mice. The reviewer wondered if the expression of TdTomato always dictate the expression of Plin2 (it sees so in Figure 2D). In figure 4, besides the cell clustering based on the expression of TdTomato, the expression of Plin2 could also be displayed. In this case, cluster 2 has higher tDTomato positive cells and shall expression more Plin2 and the DEG analysis (Figure 4G) may reflect this trend.

Regarding the tdTom-Plin2 mouse, it is important to note that this model is an endogenous knock-in of tdTomato linked to Plin2 with a linker sequence, resulting in the expression of both tdTomato and Plin2 within a single construct. Concerning Figure 4, we would like to clarify that we have indeed shown the expression of Plin2 next to the clustering of the tdTomato in the initial version of the manuscript, please see the dot-plot in figure 4 E. As Plin2 is a relatively low expressed gene, we opted for the dot-plot format to visualize these results. The dot-plot illustrates that cells positive for tdTomato also express more Plin2. Even if the tdTomato signal seems to correlate with Plin2 mRNA levels, we would like to point out again that the strength of sorting for tdTomato reflects the actual presence of LDs and not just Plin2 mRNA expression, thus allowing the detection of cells with LDs even if they would not show a change in Plin2 mRNA.

3. It is interesting that 95 % of the NSPCs expressing PLIN2 also expressed tdTom-PLIN2 and 80% of all LDs expressing PLIN2 showed colocalization with the endogenous tdTom-PLIN2. If the author used the classical immunofluorescent staining (lipid dyes OilRedO or BD493) to identified the LDs, what would be the colocalization in the proportion of tdTom-PLIN2?

We had previously done this analysis (Ramosaj and Madsen et al, Nature Communications 2021) showing that PLIN2 and BD493 were positive correlated. Now, we have repeated this analysis using endogenously tagged Plin2 using tdTom-Plin2 NSPCs, BD493 and two other lipid dyes (see below). Our results indicate that depending on the dye used, the colocalization is around 90%. Please also refer to the general part of this point-by-point reply about the new protocol to visualize LDs.

Figure removed here for copy-right issues, it can be found in the manuscript Petrelli, Rey et al, BioRxiv 2024

Redacted

4. The authors found some DEGs in different clusters of microglia (Figure 4C,F,G), what are their functional relevance? Are there some key factors to regulate the transformation from different microglia states?

We have indeed identified DEGs among the different clusters of microglia, and have tried to gain deeper insights into their significance through gene enrichment analysis (Figure 4 H). The GO terms such as “*regulation of cell activation*”, “*positive regulation of cell motility*”, “*immune effector process*” and others show that the microglia containing LDs seem to be in a certain state of activation. However, we cannot make a statement from a gene expression signature to a functional role. While the signature shares similarities with previously described signatures like DAM or LDAM, (see suppl. Figure 4) we cannot conclusively determine the functional relevance of these gene expression alterations. We would need functional assays to understand whether the LD containing microglia in the healthy brain show any functional alterations. We will explore this in future studies.

5. In Figure 6 the data showed minor differences and the conclusion “Short-term high fat diet increases the total TAG levels in the brain” is not convincing. Besides, classic immunofluorescent staining lipid dyes BD493 in SD and HFD should be added to compare the LD changes in whole mounts lateral ventricle wall.

We agree with the Reviewer that the effects are small. We still find it notable though that 4 weeks of a HFD already lead to detectable differences in TAG levels, measured by lipidomics and in LD coverage measured with the tdTom-Plin2 reporter mouse. As we have now solid proof with classical staining methods that the tdTom-Plin2 positive LDs do indeed also reflect true LDs in vivo and in vitro (please see the general part of the point-by-point reply and the detailed reply to Reviewer 1), we don't think additional staining would change this picture. In the revised version, we have toned down the statement about the LD changes with HFD, pointing out that the differences are small. We have added a statement in the discussion that a long-term HFD would need to be performed to see if LDs and TAGs would show a bigger increase.

Minor Concerns:

1. The schematic illustration of genetic process in Figure 1A is not clear to understand the generation of tdTom-Plin2 mouse, even was referred to the methods.

We have modified the scheme to make it clearer.

2. Authors should add the quality assessment of scRNA-seq, such as detected gene number, mapping ratio and mitochondrial genes percentage.

We agree that this is an important information and we have now included the following data using the Seurat function in R: number of Unique Molecular Identifiers (UMIs), detected gene number, and mitochondrial gene percentage (revised supplementary Figure 4). The average number of UMIs detected is approximately 1747, coupled with an average gene detection of 916 genes per cell. Furthermore, our data exhibit a high mapping rate to the genome at 91%, along with a mitochondrial gene percentage of 4,2%. These metrics are indicative of good data quality, particularly considering that our dataset originates from adult mice.

Reviewer #1 (Remarks to the Author):

I want to commend the authors on their hard work to address all of my comments. Particularly by providing a wealth of new data (and a new paper to come) showing that PLIN2 indeed is present on lipid droplets. This greatly strengthens the manuscript. All other comments have been appropriately addressed. I want to reiterate that this PLIN2 mouse will be a valuable tool for the community. And showing, convincingly, that lipid droplets are present in the brain at basal states is an important finding. I am happy to recommend this study for publication.

Reviewer #2 (Remarks to the Author):

The response to reviewers was very thorough and all reviewer points were either addressed or clarified sufficiently. The proteomics data is really appreciated, as is the extensive work using the lipid dyes in conjunction with the PLIN2 reporter and the extensive histological analysis of the knock-in line. The characterization of the different lipid dyes in the brain tissue will very helpful to the community and clarified the discrepancies between this work and previous studies, we are glad that the authors will provide this work to the community in a separate manuscript.

Following review of the revised manuscript, there are a number of minor issues in the text that were noted:

Line 46: "secretes" instead of "secrets"

Line 77: sentence beginning: "While neutral..." is an incomplete sentence

Line 120: The use of the phrase "endogenous tdTom-PLIN2" is sort of confusing; we know the authors are trying to emphasize that the knock-in line is an endogenous gene tagged with tdTomato, but as a reader, one more readily thinks of the untagged version as the "endogenous" version. Especially since the authors are also referring to the untagged PLIN2 here, it might be better to refer to use the phrase "knock-in tdTom-PLIN2" here or simply just refer to it as "tdTom-PLIN2" since the reader should already understand that this is a knock-in reporter line. Same point for lines 203, 210, 273, Fig 1A, 2E, 2L, 3C, 5F, 7C, Figure S5A legends.

Line 446: Please review this sentence, it seems to be missing the word "to" in a couple of places.

Line 446: states "A very recent addition to the LD imaging toolbox is two LD reporter zebrafish models with fluorescently tagged endogenous Plin2 (48) and Plin2/Plin3 (49). Only the zebrafish lines reported in ref 49 are knocked in to the "endogenous locus". This sentence should be corrected. Once again it is the meaning of "endogenous"

Line 486: "extent" instead of "extend"

The use of lowercase and uppercase when referring to the mRNA and protein is not always consistent throughout the manuscript, for example on line 124: "the tdTom-Plin2 tag" should be tdTom-PLIN2?

Reviewer #3 (Remarks to the Author):

The authors' revise mostly satisfies me. I was amazed that commercially available LD labeling reagents such as BODIPY and LipiDye can differ significantly in their properties. I hope that the authors will clarify why these differences occur in the paper they plan to submit independently. On the other hand, considering the LD dynamics in the healthy brain, the relationship with other organelles such as mitochondria can be examined by a relatively simple method such as

immunostaining by utilizing the reporter mice developed by the authors. Therefore, the authors' response that time in the order of years is required is a bit disconcerting. The authors' newly added proteomic data also suggest that LD synthesis and degradation are active, and such LD metabolism may suggest a linkage with other organelles. In any case, this reviewer commends the authors for the precise execution of the revise, resulting in a paper with stronger supporting evidence for the authors' claims.

Reviewer #4 (Remarks to the Author):

The revised manuscript has shown significant improvement in clarity and overall quality. I recommend accepting this study for publication.

Point by point reply NCOMMS-22-29252-T

We would like to thank the reviewers for their overall appreciation of our revised manuscript and for their suggestion to accept it for publication in Nature Communications.

Reviewer #1 (Remarks to the Author):

I want to commend the authors on their hard work to address all of my comments. Particularly by providing a wealth of new data (and a new paper to come) showing that PLIN2 indeed is present on lipid droplets. This greatly strengthens the manuscript. All other comments have been appropriately addressed. I want to reiterate that this PLIN2 mouse will be a valuable tool for the community. And showing, convincingly, that lipid droplets are present in the brain at basal states is an important finding. I am happy to recommend this study for publication.

We thank this reviewer for her/his overall appreciation and for sharing the view that this mouse model has great value.

Reviewer #2 (Remarks to the Author):

The response to reviewers was very thorough and all reviewer points were either addressed or clarified sufficiently. The proteomics data is really appreciated, as is the extensive work using the lipid dyes in conjunction with the PLIN2 reporter and the extensive histological analysis of the knock-in line. The characterization of the different lipid dyes in the brain tissue will very helpful to the community and clarified the discrepancies between this work and previous studies, we are glad that the authors will provide this work to the community in a separate manuscript.

We thank this reviewer for her/his overall appreciation and for sharing the view that this work is relevant for the LD community.

Following review of the revised manuscript, there are a number of minor issues in the text that were noted:

Line 46: “secretes” instead of “secrets”

Line 77: sentence beginning: “While neutral...” is an incomplete sentence

We thank the reviewer for his/her careful reading and have corrected these sentences.

Line 120: The use of the phrase “endogenous tdTom-PLIN2” is sort of confusing; we know the authors are trying to emphasize that the knock-in line is an endogenous gene tagged with tdTomato, but as a reader, one more readily thinks of the untagged version as the “endogenous” version. Especially since the authors are also referring to the untagged PLIN2 here, it might be better to refer to use the phrase “knock-in tdTom-PLIN2” here or simply just refer to it as “tdTom-PLIN2” since the reader should already understand that this is a knock-in reporter line. Same point for lines 203, 210, 273, Fig 1A, 2E, 2L, 3C, 5F, 7C, Figure S5A legends.

We understand that the term “endogenous” seems to be confusing, we indeed used it to emphasize that it is a knock-in line. We have now removed it at the suggested places.

Line 446: Please review this sentence, it seems to be missing the word “to” in a couple of places.

We have corrected this sentence accordingly.

Line 446: states “A very recent addition to the LD imaging toolbox is two LD reporter zebrafish models with fluorescently tagged endogenous Plin2 (48) and Plin2/Plin3 (49). Only the zebrafish lines reported in ref 49 are knocked in to the “endogenous locus”. This sentence should be corrected. Once again it is the meaning of "endogenous

We thank the reviewer for pointing this out, we have corrected it accordingly.

"Line 486: “extent” instead of “extend”

This is now corrected.

The use of lowercase and uppercase when referring to the mRNA and protein is not always consistent throughout the manuscript, for example on line 124: “the tdTom-Plin2 tag” should be tdTom-PLIN2?

We thank the reviewer for this observation. We checked the manuscript for these inconsistencies and have corrected it to uppercase when we talk about the protein and to lower case when it is about the mRNA.

Reviewer #3 (Remarks to the Author):

The authors' revise mostly satisfies me. I was amazed that commercially available LD labeling reagents such as BODIPY and LipiDye can differ significantly in their properties. I hope that the authors will clarify why these differences occur in the paper they plan to submit independently. On the other hand, considering the LD dynamics in the healthy brain, the relationship with other organelles such as mitochondria can be examined by a relatively simple method such as immunostaining by utilizing the reporter mice developed by the authors. Therefore, the authors' response that time in the order of years is required is a bit disconcerting. The authors' newly added proteomic data also suggest that LD synthesis and degradation are active, and such LD metabolism may suggest a linkage with other organelles. In any case, this reviewer commends the authors for the precise execution of the revise, resulting in a paper with stronger supporting evidence for the authors' claims.

We thank the reviewer for his/her overall appreciation of the revision work. We agree that the dynamics aspect has not yet been addressed. We will work on this aspect and hope to be able to provide results regarding the interaction with other organelles in the future.

Reviewer #4 (Remarks to the Author):

The revised manuscript has shown significant improvement in clarity and overall quality. I recommend accepting this study for publication.

We thank this reviewer for her/his overall appreciation and for the suggestion to accept it for publication.